# Development of an Open-Source Testbed Based on the Modbus Protocol for Cybersecurity Analysis of Nuclear Power Plants

**Israel Barbosa de Brito** [†] and **Rafael T. de Sousa, Jr.** *,[†]

Electrical Engineering Department (ENE), University of Brasilia (UnB), Brasilia 70910-900, Brazil
* Correspondence: desousa@unb.br; Tel.: +43-660-3287490
† These authors contributed equally to this work.

**Abstract:** The possibility of cyber-attacks against critical infrastructure, and in particular nuclear power plants, has prompted several efforts by academia. Many of these works aim to capture the vulnerabilities of the industrial control systems used in these plants through computer simulations and hardware in the loop configurations. However, general results in this area are limited by the cost and diversity of existing commercial equipment and protocols, as well as by the inherent complexity of the nuclear plants. In this context, this work introduces a testbed for the study of cyber-attacks against a realistic simulation of a nuclear power plant. Our approach consists in surveying issues regarding realistic simulations of nuclear power plants and to design and experimentally validate a software testbed for the controlled analysis of cyberattacks against the simulated nuclear plant. The proposal integrates a simulated Modbus/TCP network environment containing basic industrial control elements implemented with open-source software components. We validate the proposed testbed architecture by performing and analyzing a representative cyberattack in the developed environment, thus showing the principles for the analysis of other possible cybernetic attacks.

**Keywords:** cybersecurity; nuclear power plants; Asherah Nuclear Power Plant Simulator; GNS3; PLC; OpenPLC; SCADA; ScadaBR; Modbus; ModRSsim2; industrial control systems (ICS)

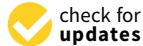

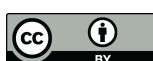

## 1. Introduction

The main question addressed by this research is the design and validation of an easily reproducible and accurate testbed for nuclear power plant (NPP) cybersecurity research. It is important to bring results regarding the protection of such cyber-physical infrastructures because there is concern of attacks against the monitoring and control systems used in real nuclear plants. However, there are inherent risks associated with the safe operation of radioactive materials and high costs involved in suspending nuclear plant operation for safely testing cyber-attacks and defense measures. This scenario makes the use of nuclear power plant simulations almost unavoidable in these situations. Therefore, presently and in the foreseeable future, this question needs to be addressed to comprehend the possible cyber-attacks, their related risks, and to compose adequate protection measures.

As lately increased computing power allows the operation of realistic simulations of nuclear reactors on personal computers, this paper contributes with the design of a robust simulation-based testbed for NPP cybersecurity studies, combining low-cost hardware and software, to enable realistic simulations of the controlled physical processes and the used communications networks. The validation of the proposal raises another paper contribution in the form of a method for simulating cyber-attacks, presenting a case scenario that illustrates how to minimize the cost, difficulty, and complexity of NPP cybersecurity analysis, while maximizing the accuracy, reproducibility, and scalability of this type of experimental setup.

Recent years have seen a rise in cases of cyber-attacks against critical infrastructure. The impact of these attacks covers a spectrum that ranges from essential service interruption and financial loss to physical destruction. These attacks have been facilitated by the

increasing digitalization in critical infrastructure sectors, and the convergence between information technology (IT) and operational technology (OT).

Examples taken from industry at large include: in 2013, the cyber-attack that paralyzed a German steel processing plant; and the attacks of 2015 and 2016 against Ukraine's power distribution grid, responsible for disrupting the power supply of thousands of homes [1]. Specifically in the nuclear industry, cases are known such as: in 2003, in the USA, the Slammer worm disabled the safety monitoring system at the Davis–Besse nuclear power plant (NPP); in 2006, in the USA, controller data traffic overload caused the shutdown of unit 3 at the Browns Ferry NPP; in 2010, in Iran, the Stuxnet worm destroyed uranium enrichment centrifuges; in 2014, in South Korea, hackers gained access to critical information about the operation of Korea Hydro & Nuclear Power NPP and demanded the shutdown of 3 reactors [2,3].

In response to the growing perceived cybersecurity threat against nuclear power plants, the academia has sought to contribute on several fronts. Research in this area encompasses proposals such as: qualitative assessments; best practice proposals; risk evaluations; cyber-attack scenario studies; development of intrusion detection systems (IDS); precautions with supply chain; and cyber–physical protection systems; among others.

Many of these studies rely on computer simulations as their main working tool, be it purely software-based, or in a hybrid configuration (hardware-in-the-loop, HIL). Indeed, for many decades, the use of computer simulations has turned into established practice in the nuclear industry, particularly for training purposes, but also in the design and licensing phases of the construction and operation of the reactors. The inherent risks associated with the safe operation of radioactive materials and the high costs involved make the use of simulations unavoidable in these situations [4].

Nuclear codes and full scope simulators are of high complexity and financial value, and are generally beyond the wider reach of the academic community. Additionally, they offer little flexibility, as they are designed specifically for the NPP model where they will be employed. Finally, they do not address important aspects for real-world cybersecurity studies, such as industrial communications networking and the interfacing of OT with the company's IT structure. This scenario leaves researchers with the task of developing appropriate testbeds in order to: on the one hand, be able to draw sufficiently general conclusions about the cybersecurity of nuclear power plants; and, on the other hand, avoid oversimplification of the simulated scenarios.

Fortunately, in recent years, computing power has increased enough to allow the operation of realistic simulations of nuclear reactors on personal computers. Examples are the series of simulators developed and made available to the public by the International Atomic Energy Agency (IAEA) [5,6]. On the other hand, operating system (OS) virtualization technology has become popular to the point of enabling, by means of virtual machines (VM), the integration of these simulations with other typical OT elements in the form of software, such as supervisory control and data acquisition system (SCADA) and programmable logic controllers (PLC); in communication networks that use protocols specific to the automation industry. Together, these techniques enable the conformation of robust testbeds for NPP cybersecurity studies.

The purpose of this paper is to take advantage of these developments to propose one such testbed. Centered on the possibility of carrying out cyber-attacks against the Asherah NPP Simulator (ANS), developed by the University of Sao Paulo (Brazil) for the International Atomic Energy Agency (IAEA) Coordinated Research Project (CRP) "Enhancing Computer Security Incident Analysis at Nuclear Facilities" [4,7]; integrated into a Modbus/TCP virtual network communicating with complementary OT elements based on open-source software.

The remainder of the paper is organized as follows: Section 2 review the literature and explore research gaps to be improved; Section 3 describes how the proposed testbed was designed and implemented; Section 4 applies the testbed to perform a specific cyber-attack scenario and evaluates the experimental results; Section 5 discusses the possibility of employing the testbed for intrusion detection and defensive capabilities research; and

Section 6 draws general conclusions, discusses limitations and suggests areas for future studies.

## 2. Related Work

Our general hypothesis argues for the potential benefits of adoption of purely software-based industrial testbeds or in combination with low-cost hardware, for cybersecurity research purposes. Furthermore, we believe that these need to be based, as far as possible, on realistic simulations of the controlled physical processes and of the communications networks used, both in its OT and IT dimensions. In this way, we will be able to minimize the cost, difficulty, and complexity while maximizing the accuracy, reproducibility, and scalability of this type of experimental setup.

From this perspective, we list below some related work (this does not focus on the uses of a HIL configuration for training purposes such as [8]); by way of example and without any attempt to exhaust the list of initiatives in the area. At the same time, we recognize that some of these works consider testbed development to be only a preliminary step to achieve diverse specific research goals. Nonetheless, we believe that the principles that guided the development of our testbed can be of value to the cybersecurity community in general.

C. N. Boldea, 2011 [9], suggested an open-source software framework to setup a SCADA testbed where the network would be provided by the application GNS3, connecting at one end a Modbus client simulator (ModRSsim2) and at the other end a SCADA server (Free Scada). The author indicates the possibility of performing DoS attacks from a VM situated in the same network against port 502 of the Modbus client. The article presents some good ideas, but does not offer further elaboration or describe practical results eventually obtained.

J. Z. Thornton, 2015 [10], designed a virtual SCADA laboratory where the physical process (gas pipeline) was modeled out of complex mathematical equations simulated by the Simulink/Matlab software [11]. This allows for a more complete study of the behavior of a physical system during a cyber-attack. The control logic expressed in ladder language was emulated by Python programming. The communication via Modbus/TCP by Python libraries (modbus_tk). The SCADA was partly implemented with a proprietary solution (GE iFix) and partly by Python libraries (TKInter). The pervasive use of Python on the testbed, while positive from a monetary perspective, could have contributed to diminishing the realism of the proposed virtual lab. Furthermore, the use of proprietary software should be avoided, if possible, in our view.

M. Andrey Teixeira et al., 2018 [12], present the development of a SCADA testbed to be used for cybersecurity research. Their setup was dedicated to controlling a water storage tank in a HIL configuration via the Modbus protocol. The effects of reconnaissance industrial network cyber-attacks on the testbed were assessed and used to train machine learning (ML) algorithms, in order to develop an automated IDS. However, the choice of the industrial subprocess to be simulated is too simple, and thus limits the possible practical applications of the model. Furthermore, it resorts to specific commercial hardware such as the Schneider PLC model M241CE40, which restricts the generality of its conclusions and complicates the reproducibility of the experiment.

S. Figueroa-Lorenzo et al., 2019 [13], in order to test their proposal to improve the security of the Modbus protocol, have built a virtual testbed, containing TCP/IP software network elements (firewall, routers and switches) made available by Cisco for use in the network simulator GNS3. Since the authors' approach was based on applying the Transport Layer Security (TLS) technique to traditional Modbus TCP/IP protocol, they assumed that the cybersecurity of the model is guaranteed by design. It then remained to evaluate possible problems arising from implementation flaws and low performance, which could be verified with the help of the arranged setup. This article shows the power and flexibility of virtualized testbeds, for exploring various aspects of cybersecurity of industrial control systems (ICS). However, it relied on proprietary software in its choice of implementation (Cisco GNS3 Appliances).

F. Zhang et al., 2019 [14], describe a testbed architecture to demonstrate a multilayered defense-in-depth-based IDS. That included an engineering workstation to run the SCADA, a data storage unit, a National Instruments cDAQ9188 Ethernet chassis, and a malicious computer running the Kali Linux OS. This setup allowed for different attacks, such as Denial of Service (DoS) and man-in-the-middle (MITM). However, the physical process representing a nuclear reactor subsystem was simulated only notionally. This limitation was remedied in a later work by the authors, as in 2020, F. Zhang et al. [15] proposed a HIL testbed built with the Asherah NPP Simulator (ANS), which is capable of a realistic simulation. It also comprises a PLC, in order to conduct false data injection attacks and collect data to ML training of a PLC process data anomaly detector. Still, it could be argued that the choice of commercial PLC (Siemens S7-1200) and proprietary software (Prosys OPC UA) contribute to prevent the widespread applicability of the proposed framework.

O. Pospisil et al., 2021 [1], summarize recent works in the area of industrial testbeds; motivated by the lack of quality real data in the quantities and features required for ML applications aimed at automating cybersecurity tasks. Although not specific to the nuclear industry, the study describes concepts and strategies common to the development of these testbeds. As choices related to the following factors: industrial processes to be studied; project category (physical, simulation, virtual, hybrid); application scenario (cybersecurity, education, functional testing, standards development); industrial communication protocols (Ethernet/IP, Profinet, EtherCAT, Modbus, Siemens S7); and levels of the automation pyramid to be modeled, according to the ANSI/ISA-95 model (ISO 62264) [16]. Specially levels L0 to L2, where: L0 deals with production processes (sensors and actuators); L1 with control (PLCs); and L2 with supervision (SCADA). The paper goes on to describe several testbeds set up in their university's laboratory for data collection purposes. The wealth of scenarios explored, however, may pose difficulties for researchers with fewer laboratory resources. In particular, in relation to testbeds assembled from a great diversity of physical equipment and proprietary protocols.

E. Aboah Boateng et al., 2022 [17], set up a testbed based on open-source software to compare the performance of the ML one-class neural network (OCNN) training algorithm on a Modbus/TCP network against previous works aimed at developing automated IDS for ICS. Those last employed one-class support vector machine (OCSVM) and isolation forest (IF) ML algorithms to detect PLC operation anomalies. The authors made the fortunate decision to implement the traffic light operation program; originally developed for the Siemens S&-1212C PLCs, in the open-source soft PLC OpenPLC instead. The human-machine interface (HMI) was also chosen to be provided by the free supervisory system ScadaBR. Despite this, we consider the simplicity employed for the network, consisting only of Modbus communication occurring between the HMI and the Soft PLC, to be overly limiting. This could explain why the anomalous scenarios described in the article were not really emulated, but only imagined. Moreover, the physical process of low complexity controlled by exclusively binary variables would hardly occur in real situations involving critical industrial subsystems.

In contrast to the works listed above, our proposal is intended to be both close to industrial practice and financially effective. We resort to a complex simulation of a nuclear power plant. This is in turn monitored and controlled by a supervisory system and PLC, based on opensource software and low-cost microcontroller, actually used for factory operations and remote monitoring by certain companies. The emulated network environment allows both the reproduction of communications by the popular industrial protocol Modbus, and the reproduction of cyber-attacks actually developed to be used against real ICS.

## 3. Proposed Testbed for Cybersecurity Analysis of Nuclear Power Plants

In this section, we present the requirements considered in building up the proposed testbed, and we discuss the idea that guided the validation of our design. Then, the testbed components are detailed and discussed. The followed methodological process can be seen in the flowchart depicted in Figure 1.

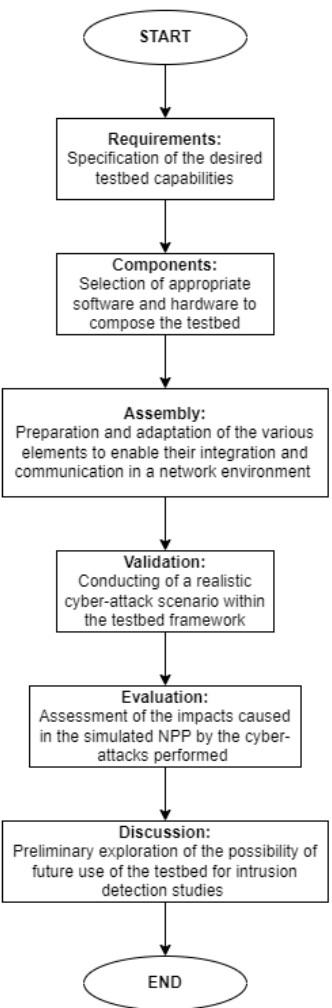

**Figure 1.** Methodological flowchart.

The requirements that guided the assembly of our testbed were as follows:

- Choice of a NPP simulator faithful to the physical processes associated with its operation;
- Use of the Modbus/TCP protocol;
- Employment of a realistic network simulator;
- Selection of open-source software for the OT and IT elements to be incorporated;
- Have the ability to perform cyber-attacks against the testbed elements;
- Having the capability to monitor and log events to record historical data.

We consider this testbed capable of emulating a variety of cyber-attacks against Modbus/TCP-based nuclear power plant simulated ICS. In order to validate this hypothesis, we planned to use it to perform an insider cyber-attack. This consisted in simultaneously: (a) replacing a local PLC by a Rogue PLC (Level 1 of the ANSI/ISA-95 model [16]), to modify the values of the registers used to control an actuator critical to the NPP operation (Level 0); and (b) interpose and modify the communication between Levels 1 and 2, so that the SCADA shows a normal situation in relation to the physical process, effectively blinding the system to the attack in progress (men-in-the-middle attack or MITM). The described levels and their respective roles can be seen in Figure 2.

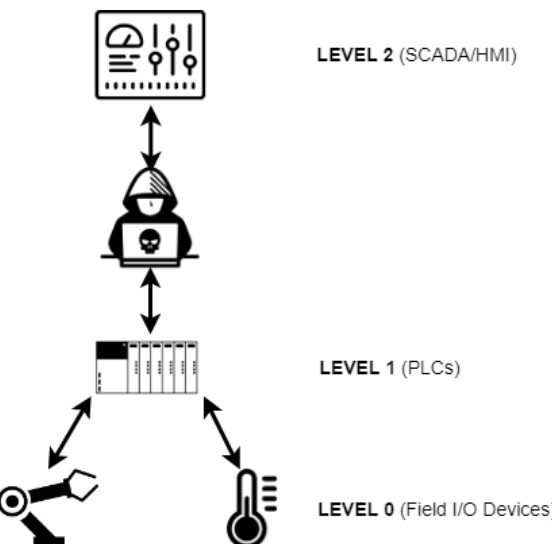

**Figure 2.** MITM Attack against the nuclear testbed.

This testbed also allows the application of the IAEA concepts of defensive computer security architecture (DCSA) [18,19] (not part of this study), a practical technique to protect facility functions that support safety and security that make use of, depend on, or are supported by digital technologies. Therefore, a researcher can implement the concepts of computer security levels (implementing a graded approach) and computer security zones (delivering defense in depth) and develop cyber-attack scenarios to assess ways in which an adversary could exploit vulnerabilities in systems performing facility functions.

The requirements and envisioned cyber-attack in turn guided the choice of the components shown in Table 1 below.

**Table 1.** Nuclear testbed. List of components.

| Role Performed | Component |
| --- | --- |
| NPP Simulator | Asherah NPP Simulator (ANS) [1] |
| Communications Protocol | Modbus/TCP |
| Modbus Simulator | ModRSsim2 [20] |
| Network Simulator | GNS3 [2] [21] |
| Software Router | VyOS (GNS3 Appliance) [22] |
| Software PLC and Ladder Program Editor | OpenPLC 1.3 (Editor and Runtime) [3] [23] |
| Arduino PLC | ESP8266 NodeMCU v1.0 ESP-12E |
| SCADA/HMI | ScadaBR 1.2 [24] |
| Cyber-attack Plataform | Kali Linux [25] |
| MITM Tool | Ettercap [26] |
| Historian | MySQL Workbench [4] [27] |
| Network protocol analyzer | Wireshark [28] |

[1] Note: running inside Simulink/MATLAB on a Windows 10 (64-bit) VM., [2] Note: VMware Workstation Player [29] and Oracle VirtualBox [30] used as hypervisors for the GNS3 appliances and VMs., [3] Note: installed on an Ubuntu 20.04 VM., [4] Note: visual tool to manage the open-source MySQL Community Edition [31] database, utilized by ScadaBR.

In what follows, we briefly explain the capabilities and justify the selection of the above listed components. We also describe the adjustments made in order to integrate them into the testbed and enable the proposed cyber-attack.

### 3.1. Modbus/TCP Protocol and the Modbus Simulator

Modbus is a protocol for industrial communications that was created more than 40 years ago (by PLC manufacturer Modicon, now Schneider Electric) and still enjoys great popularity for real world SCADA/ICS implementations. There are several reasons for

this: it is an open standard that is easy to implement and optionally available for almost all commercial automation equipment. This allows the same plant to easily integrate equipment from different manufacturers into its operations.

The protocol is also a favorite for cybersecurity studies, since in its standard form it has no mechanisms to ensure confidentiality or data integrity, among other vulnerabilities. It is also possible use search engines for Internet connected devices, like Shodan [32], to locate and remotely attack Modbus systems. Furthermore, since different brands of PLC accept the protocol as an option and it responds to external commands regardless of authentication, they can easily be victimized by injection attacks [33,34].

Last but not least, the open-source software chosen to build our testbed; specifically, ScadaBR and OpenPLC, both support the Modbus protocol. It should be noted that commercial PLC brands in general feature their own proprietary protocols and in some cases accompanying simulation software, also proprietary.

Modbus/TCP is one of three variants of the protocol and allows communication over Ethernet networks on standard port 502 (the other two being Modbus ASCII and Modbus RTU). It uses the client-server architecture and its communications are based on exchanging Ethernet Request and Response frames, as can be seen in Figure 3.

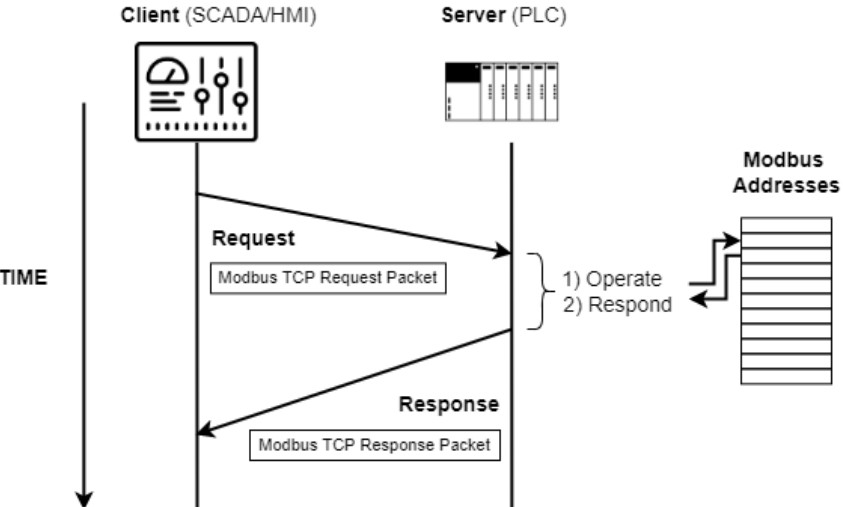

**Figure 3.** Modbus/TCP client-server communication.

Figure 4 shows how the Modbus TCP packet is encapsulated in the data section of the conventional Ethernet Frame. It is structured as follows: MBAP (Modbus Application Protocol) header followed by the PDU (Protocol Data Unit) section. This last section contains the message itself, consisting of: (a) function code, which indicates the desired operation (such as read and write); and (b) data, related to the operation defined in the previous field, such as addresses or values to write. Table 2 shows common Modbus function codes.

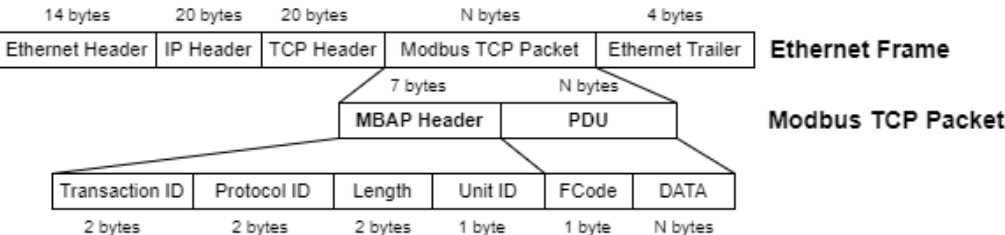

**Figure 4.** Modbus TCP packet structure.

**Table 2.** Common Modbus function codes.

| Function Code (Decimal) | Function Code (Hexadecimal) | Description |
|---|---|---|
| 01 | $0 \times 01$ | Read Coil Status |
| 02 | $0 \times 02$ | Read Input Status |
| 03 | $0 \times 03$ | Read Holding Registers |
| 04 | $0 \times 04$ | Read Input Registers |
| 05 | $0 \times 05$ | Write Single Coil |
| 06 | $0 \times 06$ | Write Single Register |
| 15 | $0 \times 0F$ | Write Multiple Coils |
| 16 | $0 \times 10$ | Write Multiple Registers |

The protocol has a particular addressing scheme that consists in dividing its memory area into four sections; for discrete (Boolean) and analog values (Coils and Registers), read-only or read-write, as can be seen in Table 3. Each of these addresses can store data types of up to 16 bits. Therefore, to use 32-bit data types, it is necessary to use two consecutive addresses for each of these values. In the particular implementation of our testbed, characterized by simulated physical processes of mainly continuous nature, we chose to use only the Holding Registers section of Modbus memory. There, all values simulated by the ANS, Boolean or analog, were saved in FLOAT 32-bit format.

**Table 3.** Modbus addressing scheme.

| Section Designation | Read | Write | Address Range |
|---|---|---|---|
| Coils | YES | YES | 00001–09999 |
| Discrete Inputs | YES | NO | 10001–19999 |
| Input Registers | YES | NO | 30001–39999 |
| Holding Registers | YES | YES | 40001–49999 |

The Modbus simulator chosen to enable communication between the different modules of the testbed was ModRSsim2 [20]. This program behaves as a server that responds to requests from Modbus clients located at different IP addresses; through port 502 of the VM where it is installed. Thus, it was used as the ANS's Modbus memory, which could then be remotely accessed by ScadaBR and OpenPLC.

### 3.2. Asherah NPP Simulator (ANS) and Its Adapted Modbus Communications Interface

The Asherah NPP Simulator (ANS) was specially developed for cybersecurity assessments, by the University of Sao Paulo, Brazil [4,7]; in the framework of an international cooperation project sponsored by the International Atomic Energy Agency (IAEA). It has a core design that mathematically simulates the nuclear physics of the Three Mile Island (TMI) reactor, the 2772 MWt Pressurized Water Reactor (PWR) Babcock and Wilcox (B&W). In addition to the core, it also simulates the various instrumentation and control (IC) modules required to operate the several subsystems commonly found in a real nuclear power plant [4].

Its unique features and the absence of similar research software availability determined the choice of ANS for our testbed. The high degree of complexity and realism of ANS could only be achieved by the developing work [35,36], and the verification and validation activities [4] performed by the developers with the support of experts involved in the IAEA CRP Enhancing Computer Security Incident Analysis at Nuclear Facilities. The proprietary software Simulink/MATLAB provided the adequate environment for the development of the ANS' 3200 blocks and more than 250 scripts. Simulink/MATLAB is the one of the two proprietary software used in our setup (the other being the OS Windows 10). In spite of this, the Simulink/MATLAB software is widely available in university laboratories around the world. ANS itself can be provided to IAEA member states upon formal request at [6]. In its current version, ANS can be deployed without the need of Matlab/Simulink, as a

runtime standalone version or in a docker/container [37] application (also without the need of any proprietary software).

In Figure 5, we can see a general view of the ANS subsystems, divided into three subsections. The two above, from left to right, show: (a) the control loops and protection system; and (b) primary and secondary loops. The bottom subsection shows the external interface, comprised mainly by the Comm Module and Matlab Historian.

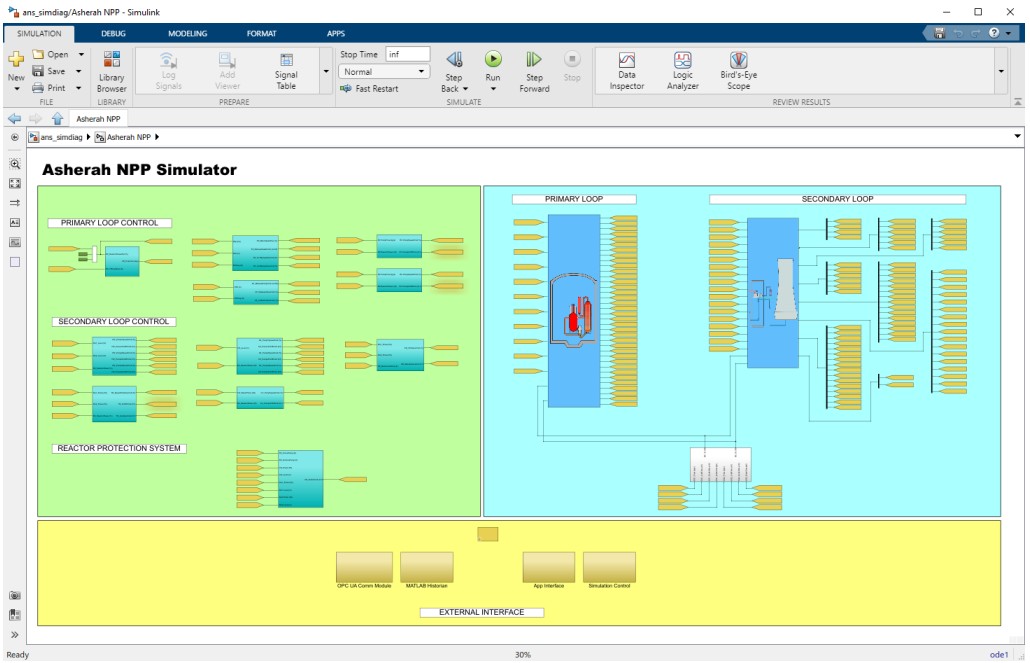

**Figure 5.** General View of the ANS subsystems.

In the version used (Windows—Release 14 dec 20), ANS was specially prepared to communicate via the OPC UA protocol (Open Platform Communications Unified Architecture). Thus, in order to enable a new Modbus communications interface, it was necessary to implement the following superficial modifications to the program: (a) map the sensor and controller variables to new areas of the Modbus server ModRSsim2 (Holding Registers Area only—two registers for each variable since they must be written as 32-bit FLOAT), as can be seen in Figure 6; (b) create a new initialization script for the new Modbus command variables (ans_load), which must be run in the Matlab Command Window; and (c) disable the original OPC UA communication modules and create and activate new Modbus modules by means of scripts based on Modbus read and write functions from MATLAB Instrument Control Toolbox, as shown in Figure 7.

Note that both applications, the Modbus server and the ANS, were installed on the same machine and therefore shared the same IP. The operating system used was Windows 10. From this point on, we could have chosen to install the rest of the testbed components on other physical machines connected by a hardware switch. Instead, we elected to build the entire testbed on a single machine by means of OS virtualization technology.

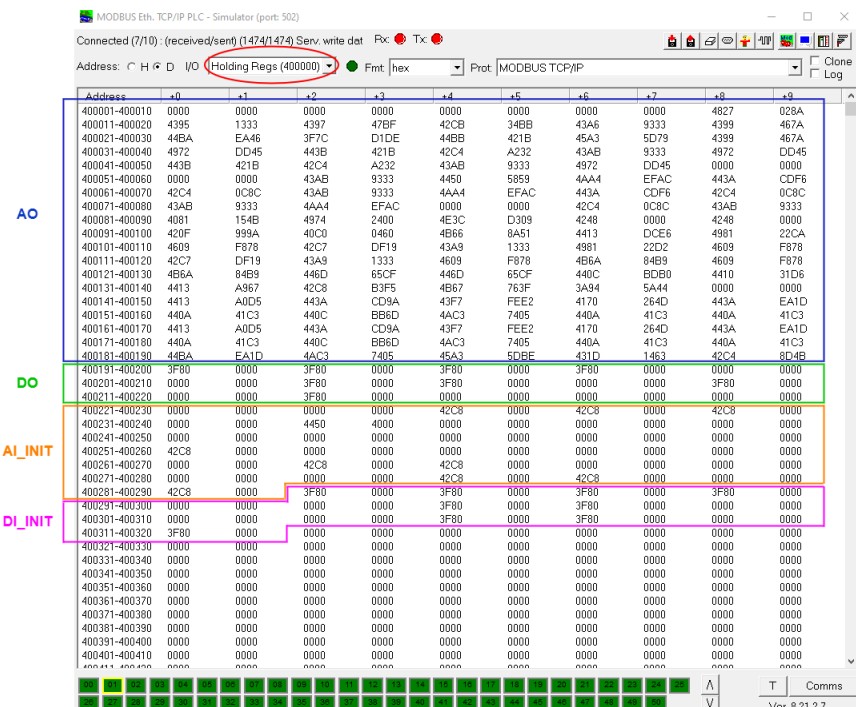

**Figure 6.** Mapped holding registers area of ModRSsim2.

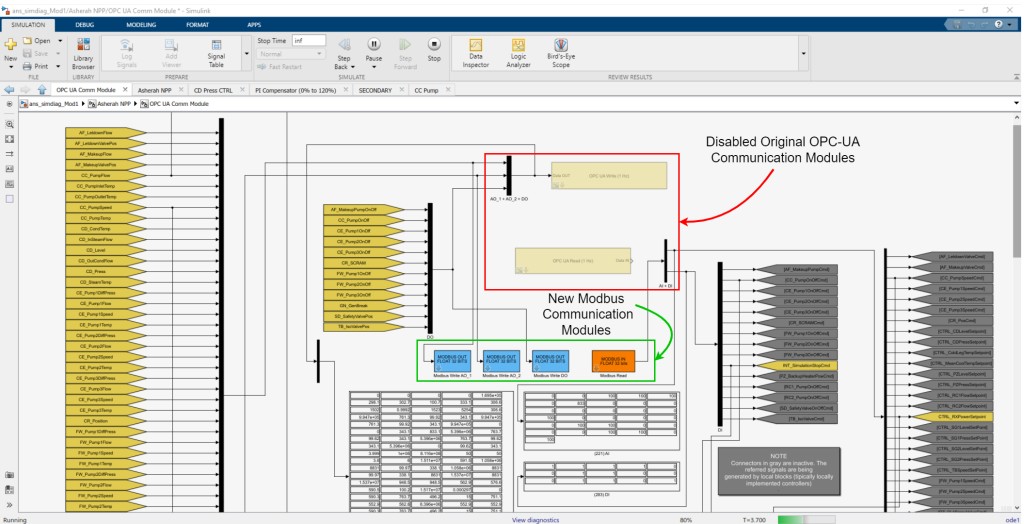

**Figure 7.** ANS Modbus communications interface.

### 3.3. GNS3 Topology

The open-source program GNS3 (Graphical Network Simulator-3) [21] was used to create a simple TCP/IP network topology needed to set up the testbed and carry out the planned insider cyber-attack. This program allows emulation and simulation of various network equipment, such as routers, switches, and firewalls; besides OS VMs. It supports several free hypervisors such as VirtualBox and VMware Workstation Player. The elements used to build the topologies are called appliances. The main elements used in our implementation, all free, were the following: VyOS Router; 2 Ubuntu 20.04 VMs (ScadaBR and OpenPLC), and; Kali Linux. Another facility provided by GNS3 is its simple integration with the well-known communication protocol analyzer software Wireshark [28], which in turn is able to analyze Modbus traffic. Several such units can be inserted into the topology segments at the same time.

In essence, our testbed was assembled to study and manipulate the Modbus/TCP communication in an industrial subnet between the PLCs of a nuclear power plant and its

supervisory system. As intervening elements, inserted in the system by the Insider, we have: (a) a computer with the OS Kali Linux distribution installed, a well-known platform used for pen test (penetration testing) and equipped with several libraries for cyberattacks; and (b) a "Rogue" PLC whose function is to substitute an internal control of the plant, previously neutralized by the malicious agent. All mentioned elements were installed in VMs whose IP and MAC addresses were fixed, and are in turn interconnected through Ethernet via a simple switch.

This configuration is sufficient to carry out insider attacks between the ANS and the HMI, since it is assumed that the subnet is segregated from the Internet in critical infrastructures such as NPP (Air Gap). However, the VyOS router [22] was also added and configured in the topology to allow access to the internet of the test environment and thus facilitate the installation and configuration of the programs and also allow for HIL testing (Arduino Wi-Fi). The resulting GNS3 topology is shown in Figure 8 and its IP scheme in Table 4 below.

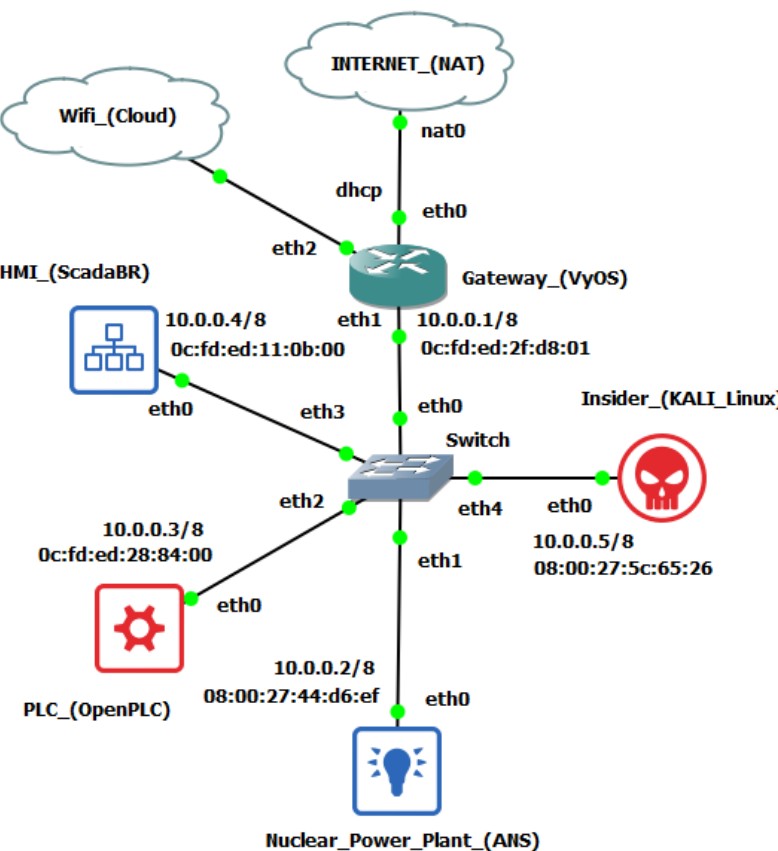

**Figure 8.** GNS3 nuclear testbed topology.

**Table 4.** Nuclear testbed IP addressing.

| Roles | Main Applications | OS | IP | MAC |
|---|---|---|---|---|
| SCADA, Historian | ScadaBR, MSQL Workbench | Ubuntu 20.04 | 10.0.0.4/8 | 0c:fd:ed:11:0b:00 |
| NPP Simulator, Modbus Server | ANS, ModRSsim2 | Windows 10 | 10.0.0.2/8 | 08:00:27:44:d6:ef |
| Cyber-attack Plataform | Kali Linux | Debian | 10.0.0.5/8 | 08:00:27:5c:65:26 |
| "Rogue" PLC | OpenPLC | Ubuntu 20.04 | 10.0.0.3/8 | 0c:fd:ed:28:84:00 |
| Router | VyOS | GNS3 Appliance | 10.0.0.1/8 (eth1), dhcp (eth0, eth2) | 0c:fd:ed:2f:d8:01 |

### 3.4. ScadaBR HMI and Historian

ScadaBR [24] is an open-source supervisory system that presents several features expected by our testbed in order to reproduce a situation close to the industry practice in a virtual environment. In particular: visualization of automation data in real time; construction of graphical screens; and continuous recording of variable changing values in a database. This last function, also called Historian or Datalogger, is provided by the relational database management system (RDBMS) linked to the main program. It is fundamental to enable deeper studies based on the analysis of data captured over long periods of time, such as, for example, the creation of datasets for IDS automation research through the training of ML algorithms. In the version we used, the supervisory links to an Apache Derby RDBMS by default. However, most ScadaBR users migrate the application to use the MySQL [31] manager instead, which would provide performance and stability gains to the Historian in real applications. We repeated the procedure in our testbed and, in order to facilitate the manipulation of this database, we also installed the MySQL Workbench [27] graphical tool in the same VM.

Regarding the choice of variables to be monitored by ScadaBR, it is important to recognize that the version of ANS that was employed continuously provides the values of 153 different input and output (IO) variables (sensors, actuators, setpoints, and commands). These are grouped into 19 subsystems distributed among the three main circuits found in PWR reactors (primary, secondary, and tertiary). Thus, any practical study involving cyber-attacks against the ANS and evaluation of its impacts must commence by selecting a relevant subset of this universe.

Besides the desired participation of the nuclear reactor (RX) and its reactivity control variables, our choice of subsystems of ANS to be included in the ScadaBR HMI was guided by their close relationship to those that are likely to be mostly affected in the cyber-attack scenario planned as a validation test for the proposed framework. To this end, we determined that the attack would primarily involve the condenser cooling pump (CCP). This system in the tertiary circuit is responsible for controlling the balance between steam and water in the condenser (CD) linked to the output of the power plant's electrical generator driving turbine (TB). Its improper operation could, in the extreme, cause the interruption of the turbine (TB) operation, and indirectly of the nuclear reactor. Another feature that makes this subsystem attractive for an insider is that it is usually physically located outside the nuclear island.

These considerations led to the development of a HMI consisting in the numerical and graphical screens shown in Figures 9 and 10. In those, we have just linked the variables associated with the chosen subsystems: main control; condenser cooling (CC); condenser (CD); turbine (TB); and reactor (RX). Their characteristics are described in greater detail in the experimental section of this paper.

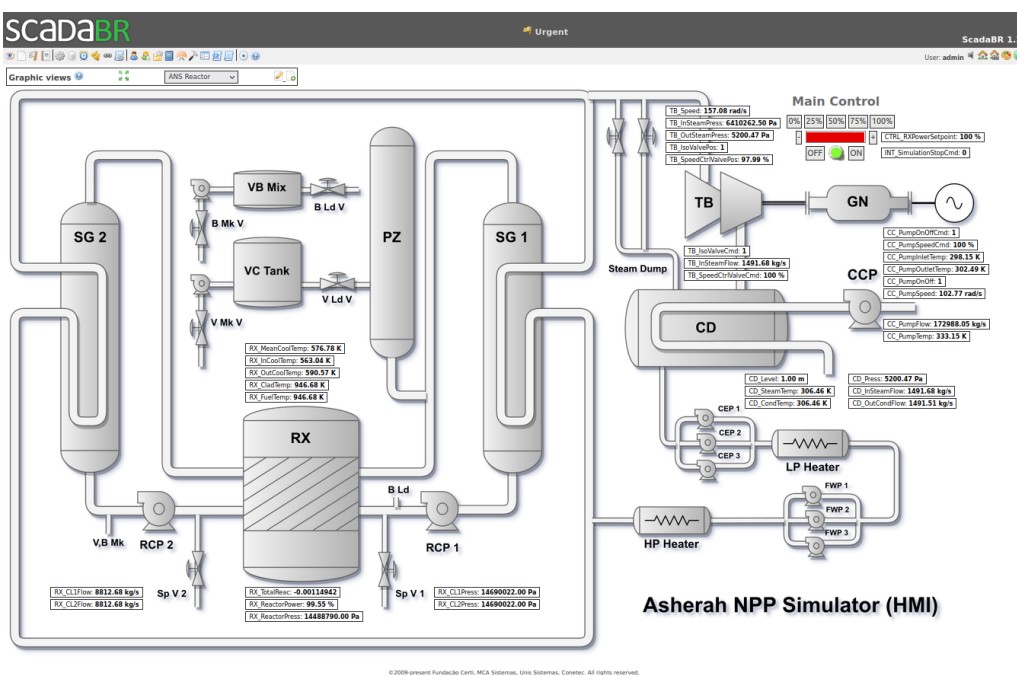

**Figure 9.** ScadaBR HMI—Synoptic Panel 1 (Numerical and Main Control).

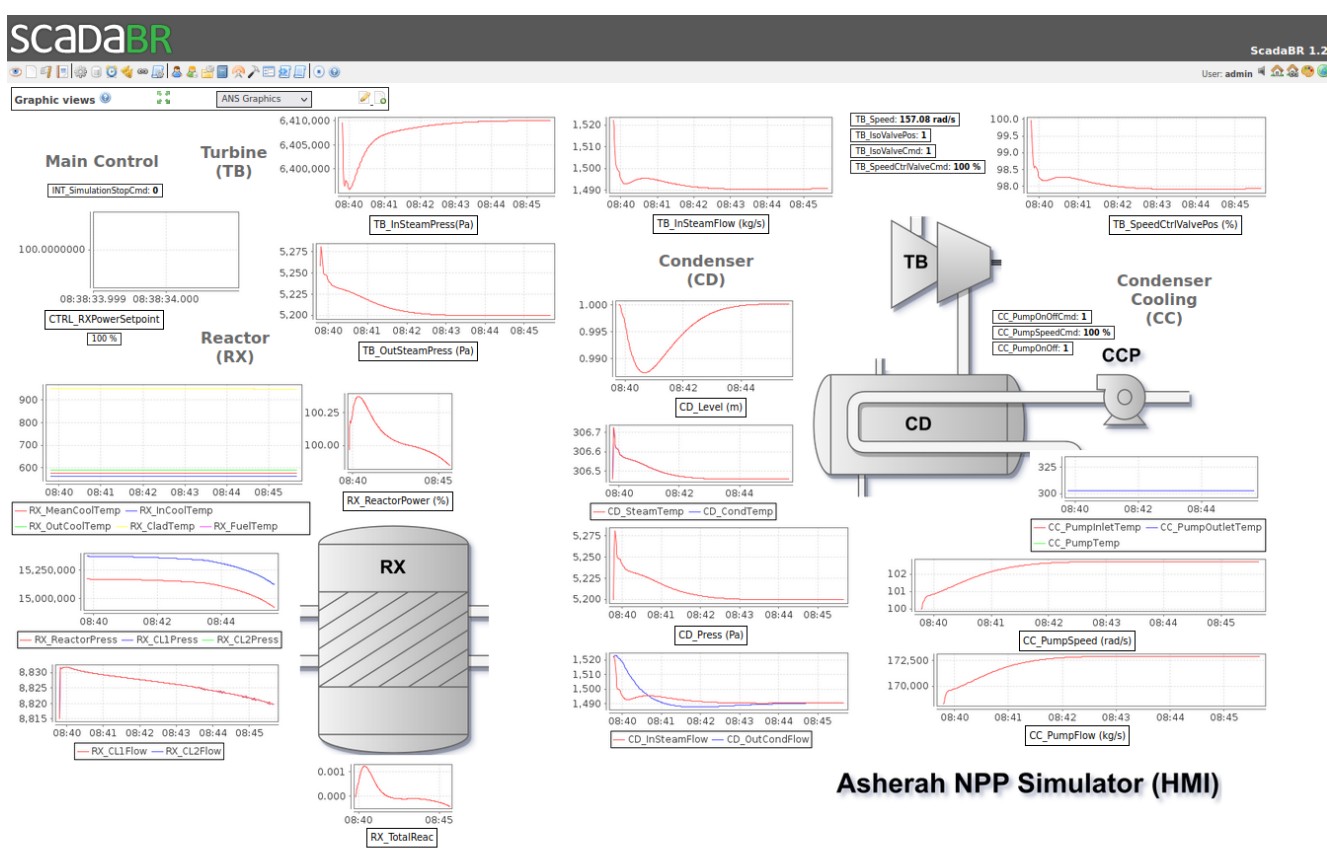

**Figure 10.** ScadaBR HMI—Synoptic Panel 2 (Graphical).

## 4. Conducting the Cyber-Attack Scenario and Evaluating the Results

As mentioned before, our cyber-attack scenario consisted in the replacement of an internal ANS control by a malicious external one that tampered with a critical variable value,

while simultaneously a real-time value-changing MITM attack prevented the anomalous activity from being detected by the HMI.

In more specific terms, the attack consisted of using the Rogue PLC to set the value of CC_PumpSpeedCmd to 75, while the HMI instead showed its normal value around 100, for main control power output of 100%. In the absence of manipulation, this variable adjusts the condenser cooling pump (CCP) speed to keep the condenser (CD) vapor pressure close to a reference value (5200 Pa), as can be seen in Figure 11.

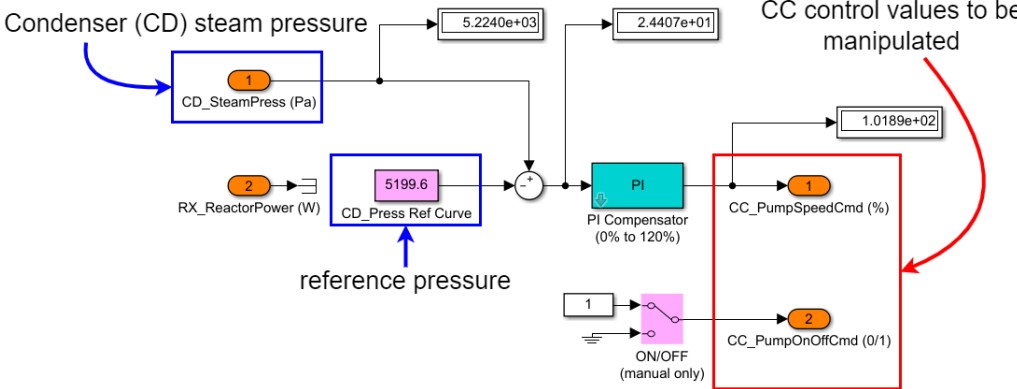

**Figure 11.** CD Press CTRL.

Thus, it was expected that the artificially imposed slightly lower fixed value would not be easily noticed and would gradually increase that condenser pressure, eventually reproducing the damaging effects described above. To do this in practice in our testbed, we followed these steps:

1.  Disabled the internal ANS control module (CD Press CTRL) that provides the value to be manipulated (CC_PumpSpeedCmd);
2.  Programmed the Rogue PLC so that it could provide the new value of CC_PumpSpeedCmd that would be externally supplied and accepted by the ANS as if it were internally generated. In addition, it was necessary to provide the control variable that keeps the pump turned on, and that was originally provided by the disabled internal module (CC_PumpOnOffCmd);
3.  Used the attacker platform to perform a MITM attack that was able to intercept and modify the Modbus/TCP communication between the ANS and ScadaBR.

### 4.1. ANS Preparation

The procedure for disabling internal ANS modules in Simulink/MATLAB involves commenting them out and at the same time enabling the necessary command connectors in the communications section. Specifically, Figure 12 shows how we disabled the CD Press CTRL module and the internal variables CC_PumpSpeedCmd and CC_PumpOnOffCmd. Figure 13 shows how we activated these same variables in the communications area, so that they can be controlled externally. The physical equivalent of this operation would be the replacement of the legitimate PLC or its control programming for another version. We assume that the malicious agent would have the ability to make this physical modification, in a real situation.

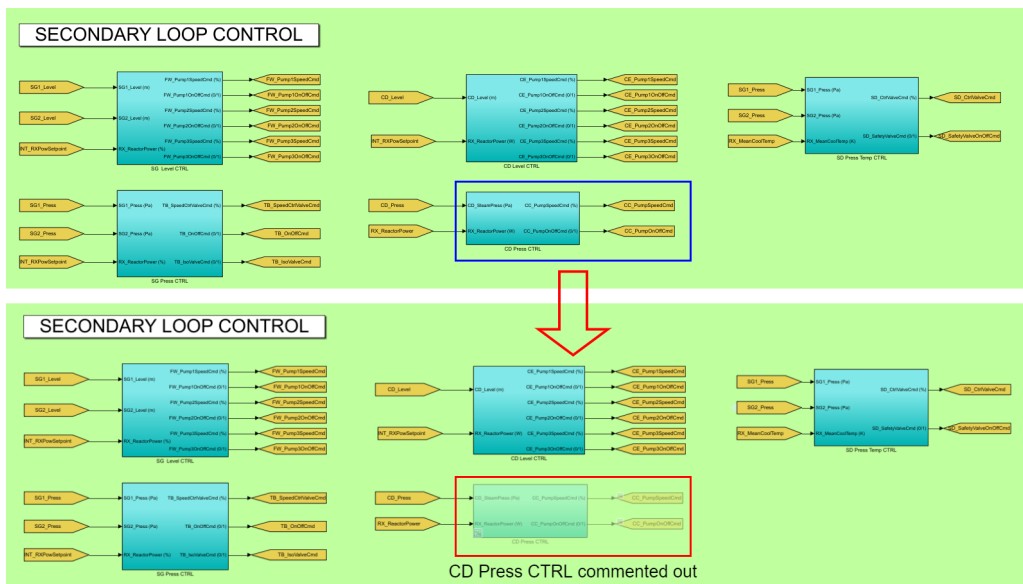

**Figure 12.** CD Press CTRL module commented out.

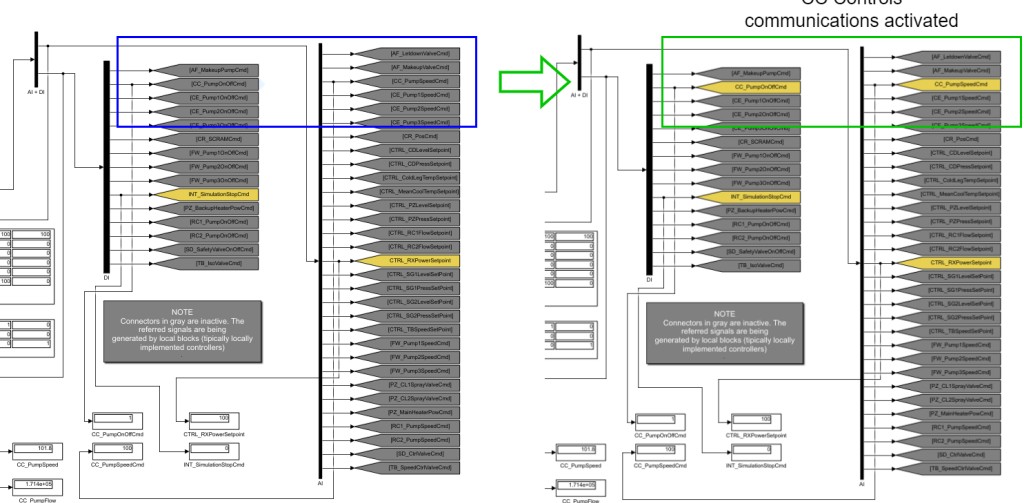

**Figure 13.** CC control variables communications activated.

### 4.2. Rogue PLC

The Rogue PLC has been implemented in the open-source program OpenPLC [23]. The software was developed according to the IEC 61131-3 standard [38], which defines the 5 PLC programming languages (Ladder Logic—LD, Structured Text—ST, Instruction List—IL, Function Block Diagram—FBD, and Sequential Function Chart—SFC). It is divided into two main parts: the Editor and the Runtime. The Editor is where programs are created. The Runtime can be embedded in low-cost microcontrollers such as the ones of the Arduino family, or in a generic target like a Soft-PLC (Windows or Linux). In addition, there is a web-based platform to define, monitor, and manage the program to be executed and the various PLCs in use.

A ladder program was designed in the OpenPLC Editor that enabled manual control of the cyber-attack, via simple circuitry based on the Arduino board ESP8266 NodeMCU v1.0 ESP-12E (button PB1 ON—attack activated; button PB2 ON—attack interrupted). This Arduino board was connected to the Wi-Fi network of the test environment [39]. To link the variables defined in the program to the Modbus memory IO target variables; to manipulate only the desired Holding Registers in it, without messing with the other addressing areas, and to have access to more analog outputs, we followed the OpenPLC addressing conventions and created 3 slaves (one of Device Type ESP8266 and two of

Device Type Generic TCP), as can be seen in Figure 14 and according to the settings shown in Table 5.

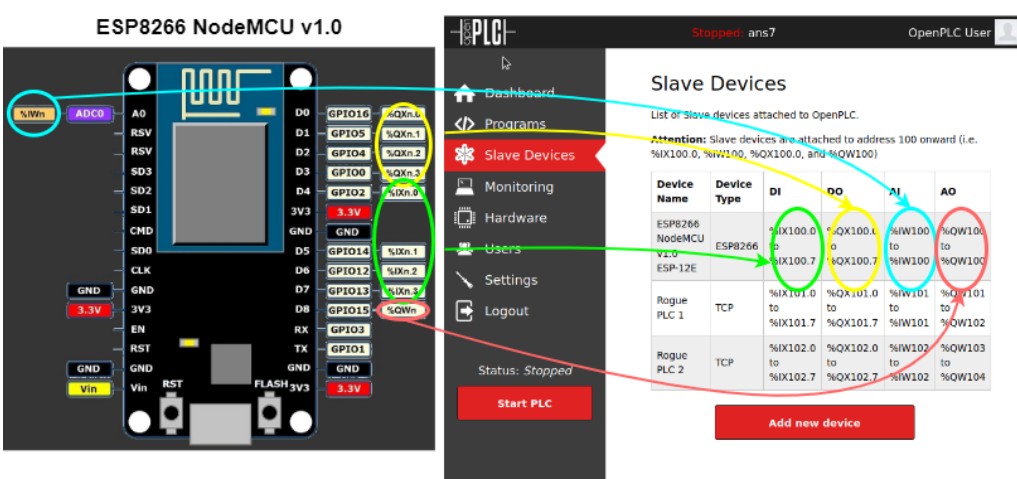

**Figure 14.** ESP8266 NodeMCU v1.0 OpenPLC address mapping.

**Table 5.** OpenPLC slave configuration parameters.

| ESP8266 NodeMCU v1.0 (physical control) | Rogue PLC 1 (CC_PumpSpeedCmd) | Rogue PLC 2 (CC_PumpOnOffCmd) |
|---|---|---|
| Device Type: ESP8266<br><br>Slave ID: 0<br>IP Address: 192.168.1.165<br>IP Port: 502 | Device Type: Generic Modbus TCP Device<br>Slave ID: 1<br>IP Address: 10.0.0.2<br>IP Port: 502 | Device Type: Generic Modbus TCP Device<br>Slave ID: 2<br>IP Address: 10.0.0.2<br>IP Port: 502 |
| Discrete Inputs (%IX100.0) Start Address: 0 Size: 8 | Discrete Inputs (%IX100.0) Start Address: 0 Size: 8 | Discrete Inputs (%IX100.0) Start Address: 0 Size: 8 |
| Coils (%QX100.0) Start Address: 0 Size: 8 | Coils (%QX100.0) Start Address: 0 Size: 8 | Coils (%QX100.0) Start Address: 0 Size: 8 |
| Input Registers (%IW100) Start Address: 0 Size: 1 | Input Registers (%IW100) Start Address: 0 Size: 1 | Input Registers (%IW100) Start Address: 0 Size: 1 |
| Holding Registers—Read (%IW100) Start Address: 0 Size: 0 | Holding Registers-Read (%IW100) Start Address: 0 Size: 0 | Holding Registers—Read (%IW100) Start Address: 0 Size: 0 |
| Holding Registers—Write (%Q100) Start Address: 00 Size: 1 | Holding Registers—Write (%Q100) Start Address: 224 [1] Size: 2 | Holding Registers—Write (%Q100) Start Address: 284 [1] Size: 2 |

[1] Note: these are the offsets pointing to the beginning of the variables in the Holding Registers area of ModRSsim2.

The ladder program logic is as follows. When the attack is triggered (the normally open button PB1 is physically pressed on the arduino board), the consecutive Holding Registers corresponding to CC_PumpSpeedCmd (Hccpspeedcmd at 400,225 and Lccpspeedcmd at 400,226) are written with the values 17,046 and 0000 (FLOAT 7.5E+01, or 75) on the ModRSsim2 Server. When the attack is stopped, the CC_PumpSpeedCmd registers are written with the values 17096 and 0000 (FLOAT 1.0E+02, or 100). The registers related to CC_PumpOnOffCmd (Hccponoffcmd at 400,285 and Lccponoffcmd at 400,286) are kept at 16,256 and 0000 (FLOAT 1.0E+00, or 1) throughout the operation. The MOVE instruction was used to transfer the content of the operand at input IN to the operand at output OUT when the Input EN is ON. The local variables used are shown in Table 6 and the implemented ladder program in OpenPLC Editor is shown in Figure 15.

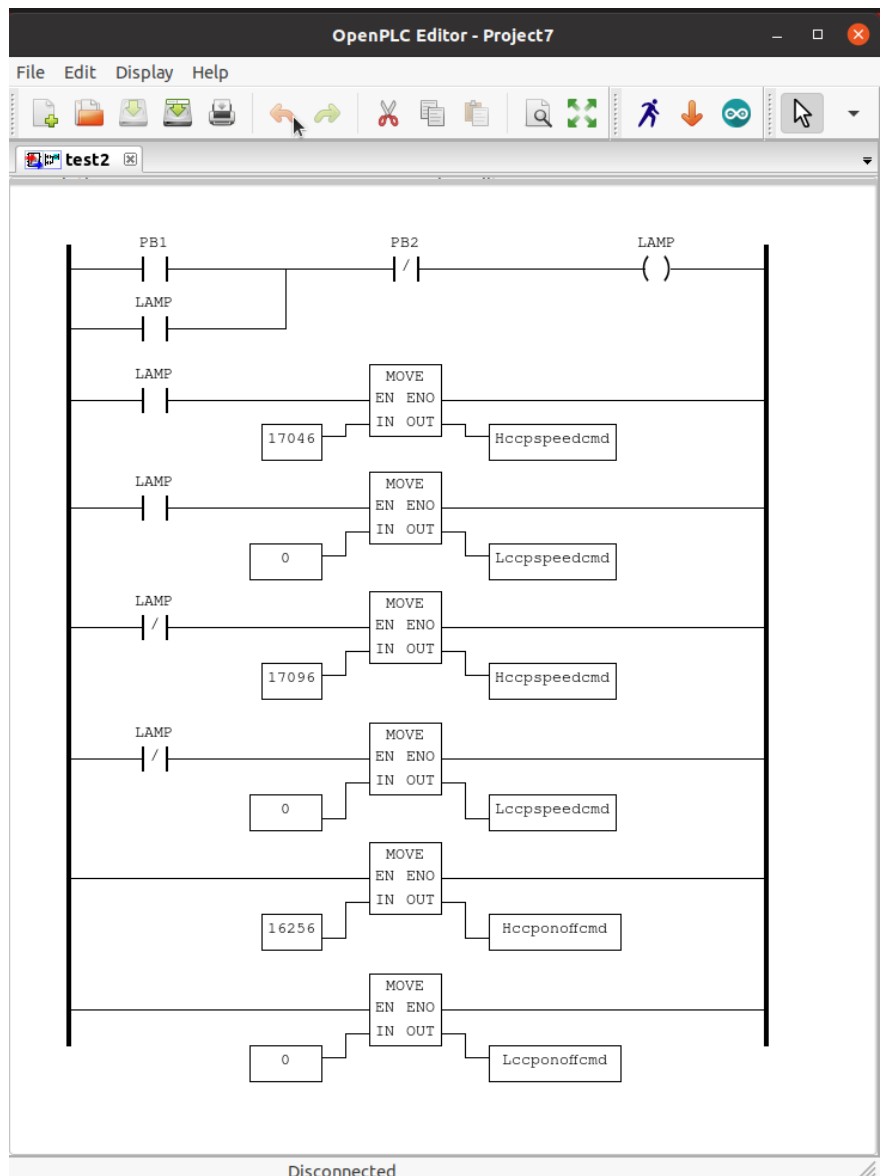

**Figure 15.** Rogue PLC ladder program in OpenPLC Editor.

**Table 6.** Rogue PLC ladder program local variables.

| Name | Class | Type | Location | Description |
|------|-------|------|----------|-------------|
| PB1 | Local | BOOL | %IX100.0 | Push button (attack ON) |
| PB2 | Local | BOOL | %IX100.1 | Push button (attack OFF) |
| LAMP | Local | BOOL | %QX100.0 | Warning LED (attack ON) |
| Hccpspeedcmd | Local | UINT | %QW101 | CC_PumpSpeedCmd (Higher Byte) |
| Lccpspeedcmd | Local | UINT | %QW102 | CC_PumpSpeedCmd (Lower Byte) |
| Hccponoffcmd | Local | UINT | %QW103 | CC_PumpOnOffCmd (Higher Byte) |
| Lccponoffcmd | Local | UINT | %QW104 | CC_PumpOnOffCmd (Lower Byte) |

### 4.3. Attack Platform

Kali Linux [25] distribution was chosen to unleash the value-changing MITM attack. This platform allows for performing cyber-attack scenarios with the purpose of assessing its consequences. It offers a wide range of tools for information security and ethical hacking-related tasks, such as penetration testing, computer forensics and reverse engineering. In its inventory there are tools tailored exclusively for cyber-attacks against SCADA/ICS. For example, the Metasploit framework, which comes pre-installed by default on Kali, offers modules that can be used to find Modbus servers and clients; and read and write

Modbus registers [40]. For some equipment, it is even possible to upload, analyze, and then download and replace the PLC ladder logic (modicon_stux_transfer module) [41].

These Metasploit modules in particular, or exploits, as they are called, could have been used in our testbed to write constant values to the ModRSSim2 server registers and thus achieve the same results as those obtained by the simple Rogue PLC logic just described. As can be seen by the sequence of commands shown in Figure 16, it is possible to employ the "modbusclient exploit" to write the values 17,046 and 0000 (value 75) for the two bytes after the 224 address offset (variable CC_PumpSpeedCmd) of the ModRSsim2 Server at IP 10.0.0.2 (ANS VM).

**Figure 16.** Metasploit Modbus injection attack example (CC_PumpSpeedCmd = 75).

Instead, we preferred to demonstrate the HIL implementation capabilities of our testbed, and emphasized its potential for increased programming complexity, and scalability provided by the use of soft PLCs; which allows the developing of cyber-attacks that require a greater knowledge of the plant control system (not addressed in this study). For example, by the external cloning the logic of the internal controller being replaced, it would be possible to enable more subtle attacks to be carried out, with greater control of the manipulated variables and eventual return to normal control whenever desired. In

principle, a well-informed Insider would know the implementation details necessary to perform this procedure. In any case, this point demonstrates the flexibility of the testbed and illustrates an alternative way to perform the same Modbus injection cyber-attack.

As for the MITM attack itself, the specific tool used was Ettercap (also present by default on Kali Linux distributions) [26]. It allows us to snoop live TCP/IP connections and to filter content on the fly. In the validation experiment we performed the in-transit change of Modbus response packets from ANS to ScadaBR. This was done in two steps. First, Ettercap applied the ARP poisoning technique (sending unsolicited ARP replies simultaneously to both of its targets) to make ANS (10.0.0.2) believe that the ScabaBR (10.0.0.4) was located in the Kali VM IP address (10.0.0.5); and to make ScabaBR believe that ANS was in Kali address, as shown in Figure 17. Ettercap could now eavesdrop and pass on these packets in both directions.

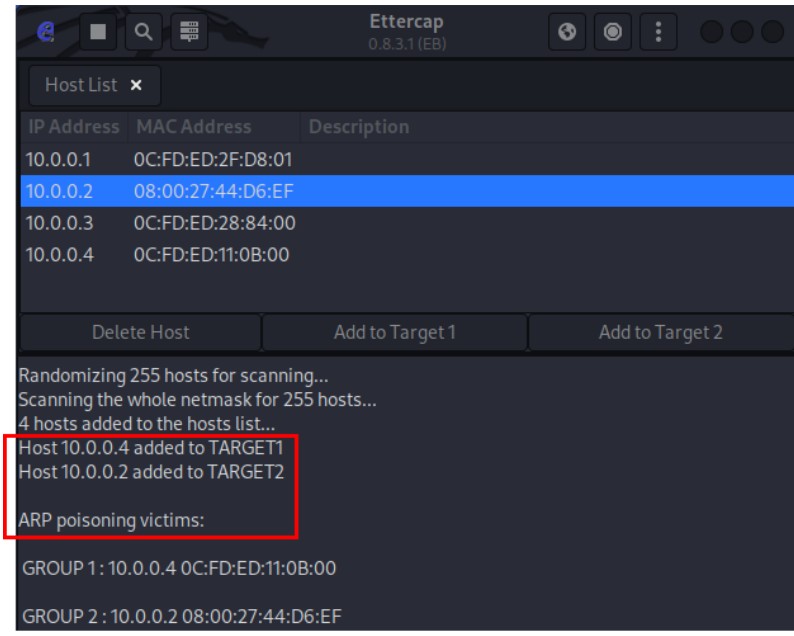

**Figure 17.** Ettercap ARP poisoning (ANS and ScadaBR).

The second step consisted in filtering and changing the ANS responses to Modbus READ requests made by ScadaBR. Among the various values contained in these response packets, we wanted to change in transit only the ones corresponding to CC_PumpSpeedCmd (to its normal value of 100, regardless of the actual value being transmitted by ANS). As Ettercap filtering was not designed with the Modbus protocol in mind, it was necessary to design a script that took into account: (a) Ettercap's filter syntax and offset addressing rules; (b) the specifics of the HMI implementation; and (c) ScadaBR's execution routines.

Ettercap filter offsets (pointed to by DATA.data + OFFSET = "value") start at the beginning of the DATA section of the Ethernet frame, which coincides with the beginning of the Modbus TCP packet section for the Modbus TCP protocol. On the other hand, for the set of variables chosen to be monitored by the HMI, ScadaBR performed 3 sequential requests whose responses had fixed length and thus could be used as identifiers (Length: 251 [ 00 fb ] Registers 8–131; Length: 243 [ 00 f3 ] Registers 132–251; Length: 113 [ 00 43 ] Registers 280–311). We also knew that we must change only the contiguous registers located at 224 and 225, to the value 0X42C80000 = 100 DEC. With these considerations taken into account, it was possible to write the appropriate filter script, shown in Figure 18 and capable of changing only the desired packets and registers.

```
################################################################################
# This Ettercap filter implements a MITM Insider Substitution Attack against ScadaBR.  #
# The chosen value is continuously shown in the supervisory, no matter what the real   #
# value read in the modbus register really is.                                         #
################################################################################

# filter [select Modbus Protocol] + [response destination ip address]

if (ip.proto == TCP && tcp.src == 502 && ip.dst == '10.0.0.4') {

# test modbus filter

   msg("response");

# condition [ScadaBR response ID]

   if (DATA.data + 4 == "\x00\xf3") {

# replace modbus registers 224 and 225 (CC_PumpSpeedCmd) for 100 dec (42C8 0000 hex FLOAT)
# regardless of their current values

      DATA.data + 193 = "\x42\xC8\x00\x00";

      msg("content of holding registers for CC_PumpSpeedCmd transmitted as 100 dec");
   }

}
```

**Figure 18.** Ettercap MITM changing values filter script.

It is understood that different configurations in the HMI would require adaptations to the presented script. Once again, we assume that the Insider has in-depth knowledge of the control architecture. It should be noted that in a real situation, we would also have several SlaveIDs for different PLCs, which in ANS correspond to internal control modules, all gathered under a single SlaveID.

### 4.4. Combined Cyber-Attack

The planned cyber-attack was successful and able to: (a) block the internal control of CC_PumpSpeedCmd; (b) inject an arbitrary constant value of 75 into CC_PumpSpeedCmd; and (c) show the normal value of CC_PumpSpeedCmd = 100 on the HMI, during the attack. Figure 19 shows the Arduino board ESP8266 NodeMCU v1.0 breadboard circuit and the OpenPLC web interface monitoring page for the implemented Rogue PLC ladder program. All the programmed variable values are displayed in real time. Additionally, while the attack is occurring, the Ettercap filter script continuously outputs the message that indicates that the MITM is in progress, as show in Figure 20. Next, in Figure 21, we can see the local value for CC_PumpSpeedCmd in the Matlab ANS interface is indeed modified to 75 by the Rogue PLC, and at the same time, the false value of 100 is presented in the ScadaBR HMI.

Figure 22 below shows another way to visualize the same cyber-attack; this time from inside the GNS3 topology with the help of two Wireshark instances. The lower one in the figure, located between the ANS VM and the Switch in the topology, captures the NPP response packets to the supervisory before the real-time modification performed by the Kali/Ettercap MITM. The upper one, between the ScadaBR VM and the Switch, captures the same packets after the modification. Since each response corresponds to the same Modbus transaction value, it is possible to check the results by reading the fields corresponding to records 224 and 225; which show 75 for the lower one and 100 for the higher one.

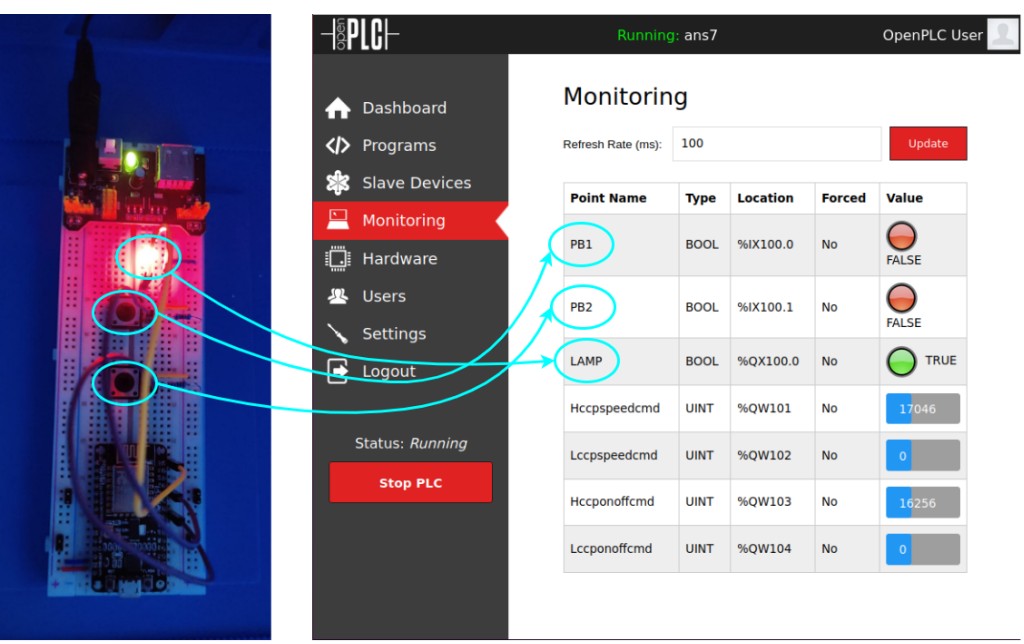

**Figure 19.** Attack ON—Rogue PLC.

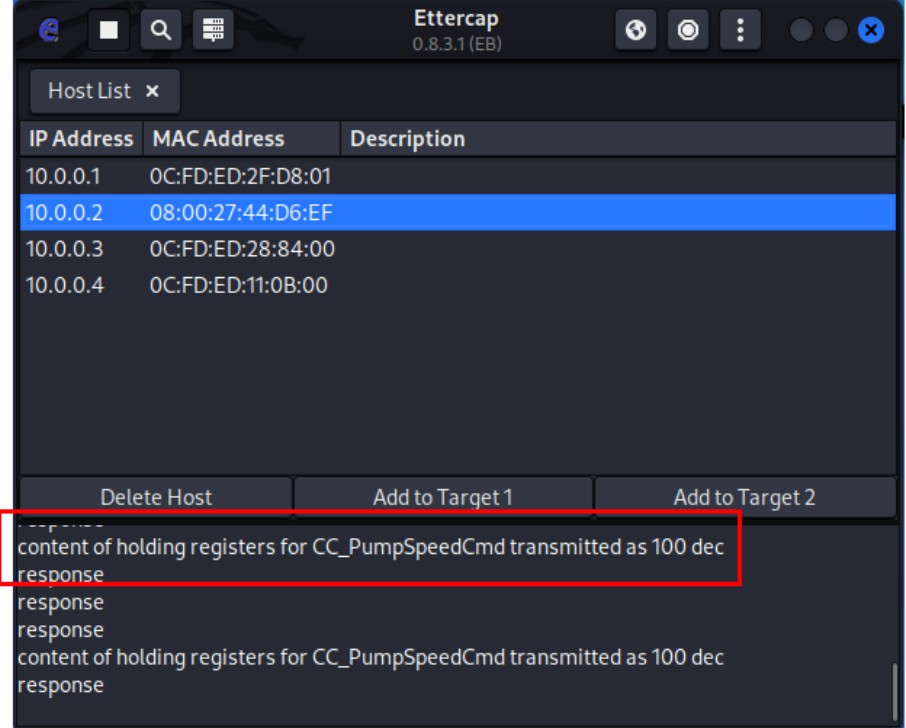

**Figure 20.** Ettercap MITM (CC_PumpSpeedCmd transmitted as 100).

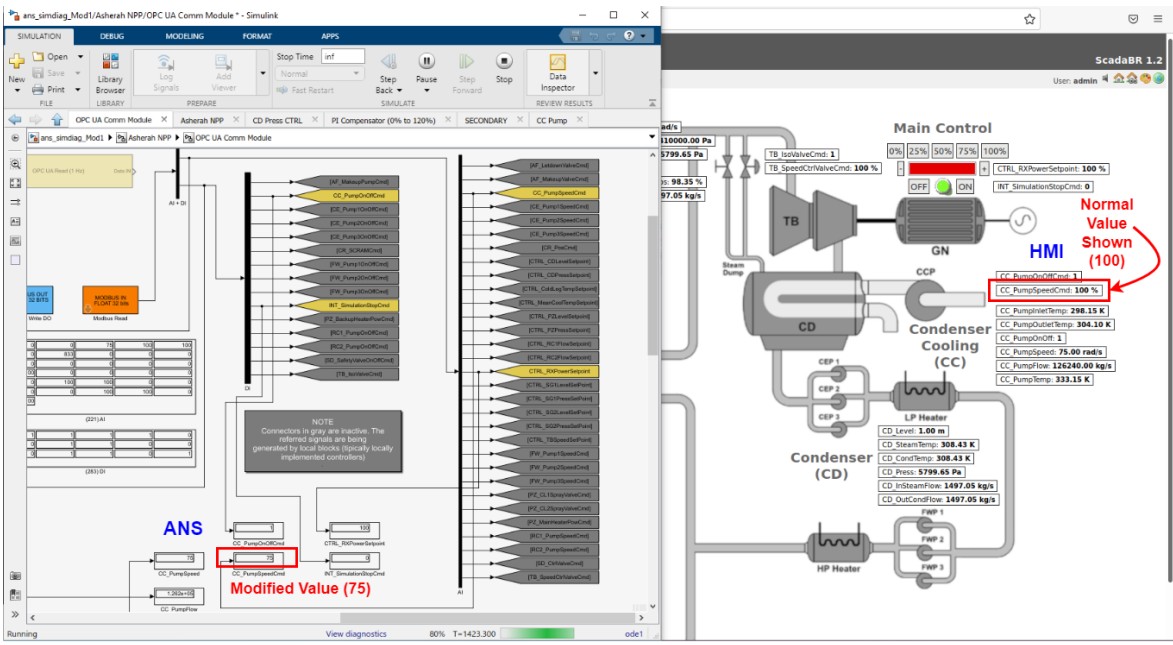

**Figure 21.** ANS and HMI under attack.

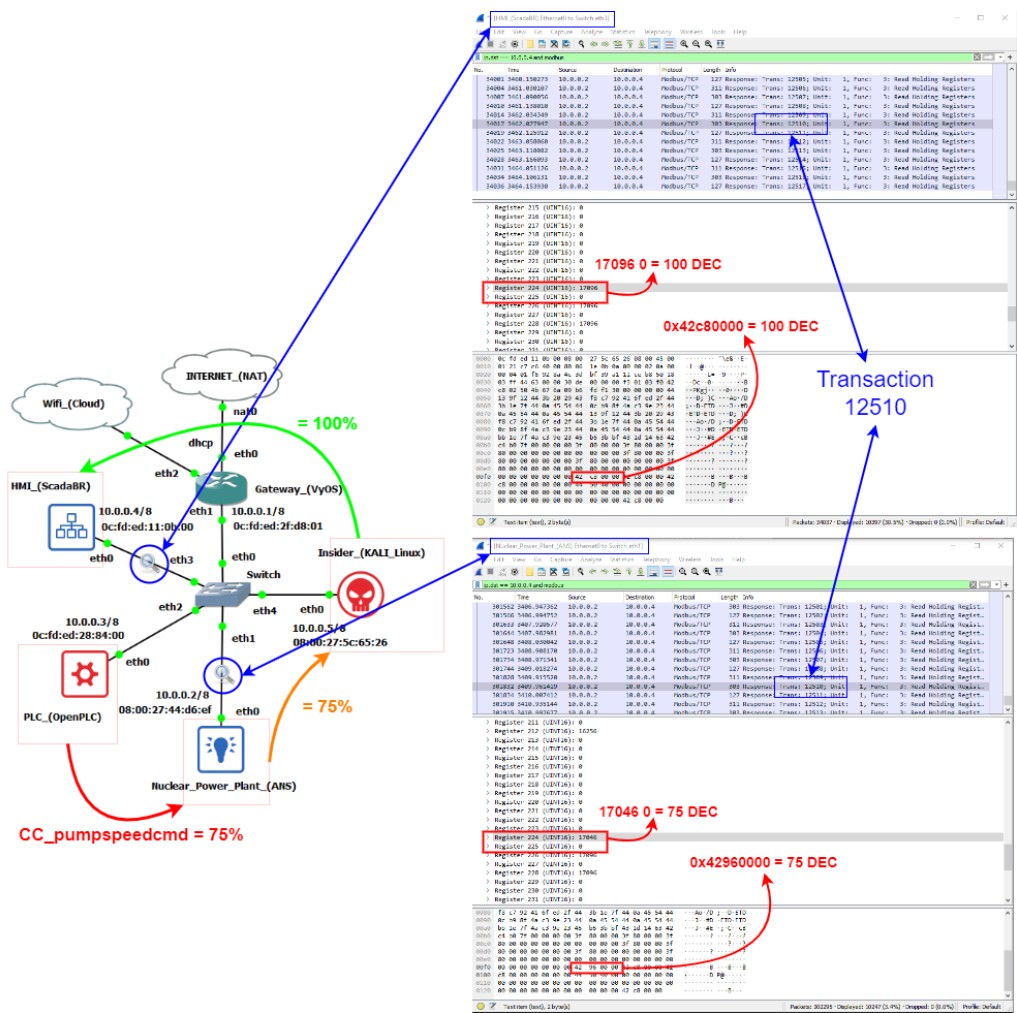

**Figure 22.** Attack with Wireshark packet analysis example.

### 4.5. Impact Assessment

To evaluate the impact of the proposed cyber-attack on ANS' subsystems, we based ourselves on the following aspects:

- Possibility of a domino effect impacting the main reactor;
- Affected variables values distance from their nominal operating values;
- The eventual triggering of the reactor protection system.

Contrary to our original expectations, setting the value of CC_PumpSpeed to 75 (through the variable CC_PumpSpeedCmd) did not significantly affect the nuclear reactor operational variables. In these circumstances, the most impacted variables were the condenser vapor pressure (CD_Press) and the turbine outlet pressure (TB_OutSteamPress). So, we repeated the attack for several different values of CC_PumpSpeed, varying it in steps of 5 units, between 75 and 15; and assessed the variables' equilibrium values in comparison with their rated values (which can be obtained from the ANS manual).

While proceeding in this way, it is important to consider the operational limits imposed by the simulator itself. The ANS has a reactor protection system (RPS) that triggers the so-called reactor´s SCRAM (emergency shutdown) whenever certain thresholds are exceeded. Figure 23 shows how its logic is implemented.

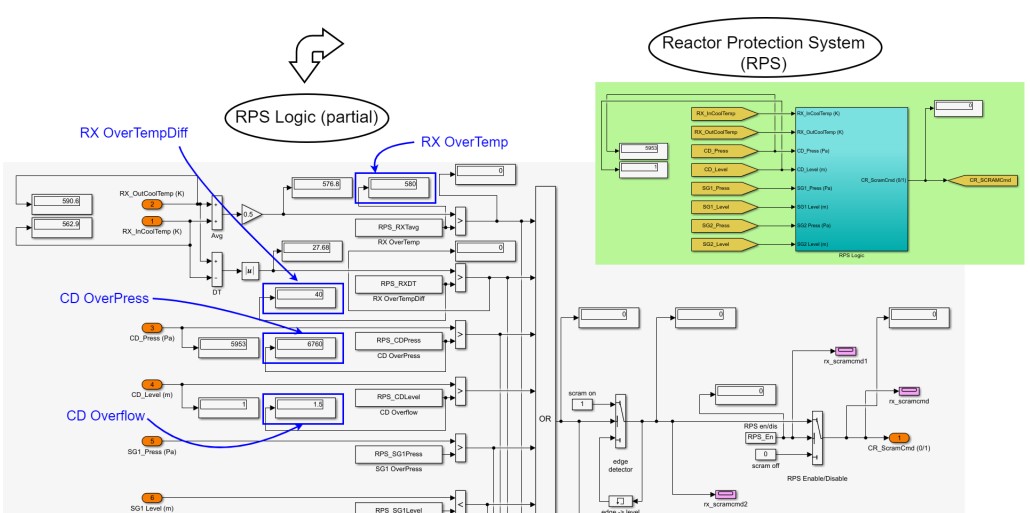

**Figure 23.** ANS reactor protection system (RPS).

For the subsystems monitored in the setup, whenever any of the following variables exceeds its respective threshold, the OR gate depicted propagates the ON (1) signal to the SCRAM Output (CR_ScramCmd): CD_Level > CD Overflow = 1.5 (m); CD_Press > CD OverPress = 6760 (Pa); mod (RX_OutCoolTemp − RX_InCoolTemp) > RX OverTempDiff = 40 (k); and 0.5 * (RX_OutCoolTemp − RX_InCoolTemp) > RX OverTemp = 580 (K). The only one of these parameters that varied for the different scenarios was the CD_Press, whose maximum threshold was reached with CC_PumpSpeed between 55 and 50. We have disabled the triggering of this protection system in order to proceed with tests for CC_PumpSpeed values below 50. However, states very far from the operating limits may lose its physical meaning for the simulation or be impossible to achieve in real equipment.

After completing these rounds of attacks, it was verified that the final results for the monitored variables at the various levels followed essentially the same pattern as revealed by the initial attack, with regard to the main variables affected; except that the condenser vapor pressure and the turbine outlet pressure increased more and more with the decrease of the condenser cooling pump speed.

In the following tables, we have applied color-coding to visually differentiate the degree of deviation of the measured values from the operating values under normal conditions. Rated values are represented in green. Values just below the rated (up to less than 10%) are in light blue, and below 10% in dark blue. Values just above the rated (up to

10%) are in orange, and above 10% in red. The tables also show other relevant information such as the original labels, ranges, units and descriptions, as defined by the ANS developers (Figures 24–27).

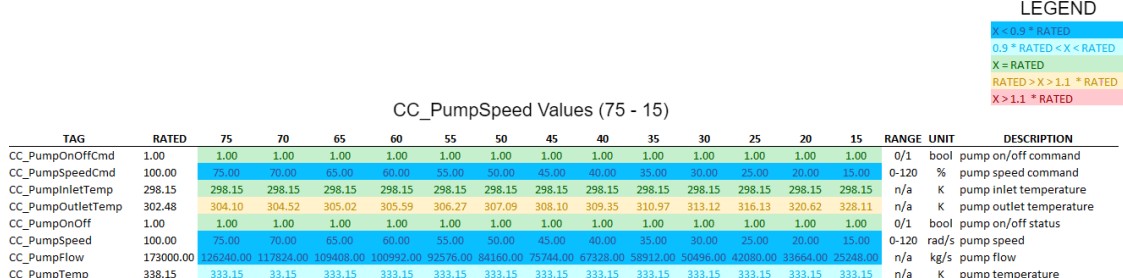

**LEGEND**

| | |
|---|---|
| X < 0.9 * RATED | (blue) |
| 0.9 * RATED < X < RATED | (cyan) |
| X = RATED | (green) |
| RATED > X > 1.1 * RATED | (orange) |
| X > 1.1 * RATED | (red) |

**CC_PumpSpeed Values (75 - 15)**

| TAG | RATED | 75 | 70 | 65 | 60 | 55 | 50 | 45 | 40 | 35 | 30 | 25 | 20 | 15 | RANGE | UNIT | DESCRIPTION |
|---|---|---|---|---|---|---|---|---|---|---|---|---|---|---|---|---|---|
| CC_PumpOnOffCmd | 1.00 | 1.00 | 1.00 | 1.00 | 1.00 | 1.00 | 1.00 | 1.00 | 1.00 | 1.00 | 1.00 | 1.00 | 1.00 | 1.00 | 0/1 | bool | pump on/off command |
| CC_PumpSpeedCmd | 100.00 | 75.00 | 70.00 | 65.00 | 60.00 | 55.00 | 50.00 | 45.00 | 40.00 | 35.00 | 30.00 | 25.00 | 20.00 | 15.00 | 0-120 | % | pump speed command |
| CC_PumpInletTemp | 298.15 | 298.15 | 298.15 | 298.15 | 298.15 | 298.15 | 298.15 | 298.15 | 298.15 | 298.15 | 298.15 | 298.15 | 298.15 | 298.15 | n/a | K | pump inlet temperature |
| CC_PumpOutletTemp | 302.48 | 304.10 | 304.52 | 305.02 | 305.59 | 306.27 | 307.09 | 308.10 | 309.35 | 310.97 | 313.12 | 316.13 | 320.62 | 328.11 | n/a | K | pump outlet temperature |
| CC_PumpOnOff | 1.00 | 1.00 | 1.00 | 1.00 | 1.00 | 1.00 | 1.00 | 1.00 | 1.00 | 1.00 | 1.00 | 1.00 | 1.00 | 1.00 | 0/1 | bool | pump on/off status |
| CC_PumpSpeed | 100.00 | 75.00 | 70.00 | 65.00 | 60.00 | 55.00 | 50.00 | 45.00 | 40.00 | 35.00 | 30.00 | 25.00 | 20.00 | 15.00 | 0-120 | rad/s | pump speed |
| CC_PumpFlow | 173000.00 | 126240.00 | 117824.00 | 109408.00 | 100992.00 | 92576.00 | 84160.00 | 75744.00 | 67328.00 | 58912.00 | 50496.00 | 42080.00 | 33664.00 | 25248.00 | n/a | kg/s | pump flow |
| CC_PumpTemp | 338.15 | 333.15 | 33.15 | 333.15 | 333.15 | 333.15 | 333.15 | 333.15 | 333.15 | 333.15 | 333.15 | 333.15 | 333.15 | 333.15 | n/a | K | pump temperature |

**Figure 24.** Simulated results (CC—Condenser Cooling).

**CC_PumpSpeed Values (75 - 15)**

| TAG | RATED | 75 | 70 | 65 | 60 | 55 | 50 | 45 | 40 | 35 | 30 | 25 | 20 | 15 | RANGE | UNIT | DESCRIPTION |
|---|---|---|---|---|---|---|---|---|---|---|---|---|---|---|---|---|---|
| CD_Level | 1.00 | 1.00 | 1.00 | 1.00 | 1.00 | 1.00 | 1.00 | 1.00 | 1.00 | 1.00 | 1.00 | 1.00 | 1.00 | 1.00 | n/a | m | Condenser level |
| CD_SteamTemp | 306.46 | 308.43 | 308.94 | 309.51 | 310.17 | 310.94 | 311.85 | 312.95 | 314.30 | 316.01 | 318.25 | 321.32 | 325.82 | 333.18 | n/a | K | Steam temperature |
| CD_CondTemp | 306.46 | 308.43 | 308.94 | 309.51 | 310.17 | 310.94 | 311.85 | 312.95 | 314.30 | 316.01 | 318.25 | 312.32 | 325.82 | 333.18 | n/a | K | Condensate temperature |
| CD_Press | 5200.00 | 5799.65 | 5952.76 | 6127.13 | 6327.79 | 6561.58 | 6919.54 | 7362.52 | 7907.64 | 8597.22 | 9693.85 | 11301.27 | 14113.60 | 19997.70 | n/a | Pa | Condenser pres sure (vacuum (absolute)) |
| CD_InSteamFlow | 1490.28 | 1497.05 | 1498.79 | 1500.78 | 1503.07 | 1505.75 | 1508.93 | 1512.78 | 1517.55 | 1523.64 | 1531.72 | 1542.87 | 1559.50 | 1587.61 | n/a | kg/s | Steam flow to condenser |
| CD_OutCondFlow | 1490.28 | 1497.05 | 1498.79 | 1500.78 | 1503.07 | 1505.75 | 1508.93 | 1512.78 | 1517.55 | 1523.64 | 1531.72 | 1542.87 | 1559.50 | 1587.61 | n/a | kg/s | Condensate flow from condenser |
| CD Overflow | 1.50 | 1.00 | 1.00 | 1.00 | 1.00 | 1.00 | 1.00 | 1.00 | 1.00 | 1.00 | 1.00 | 1.00 | 1.00 | 1.00 | n/a | n/a | Reactor Protection System |
| CD OverPress | 6760.00 | 5800.00 | 5953.00 | 6127.00 | 6328.00 | 6562.00 | 6920.00 | 7363.00 | 7908.00 | 8597.00 | 9693.85 | 11301.24 | 14110.00 | 20000.00 | n/a | n/a | Reactor Protection System |

**Figure 25.** Simulated results (CD—Condenser).

**CC_PumpSpeed Values (75 - 15)**

| TAG | RATED | 75 | 70 | 65 | 60 | 55 | 50 | 45 | 40 | 35 | 30 | 25 | 20 | 15 | RANGE | UNIT | DESCRIPTION |
|---|---|---|---|---|---|---|---|---|---|---|---|---|---|---|---|---|---|
| TB_IsoValveCmd | 1.00 | 1.00 | 1.00 | 1.00 | 1.00 | 1.00 | 1.00 | 1.00 | 1.00 | 1.00 | 1.00 | 1.00 | 1.00 | 1.00 | 0/1 | bool | Isolation Valve Command |
| TB_SpeedCtrlValveCmd | 100.00 | 100.00 | 100.00 | 100.00 | 100.00 | 100.00 | 100.00 | 100.00 | 100.00 | 100.00 | 100.00 | 100.00 | 100.00 | 100.00 | 0-100 | % | Speed control valve command (Governor Valve) |
| TB_Speed | 157.08 | 157.08 | 157.08 | 157.08 | 157.08 | 157.08 | 157.08 | 157.08 | 157.08 | 157.08 | 157.08 | 157.08 | 157.08 | 157.08 | n/a | rad/s | Turbine speed |
| TB_InSteamPress | 6410000.00 | 6410000.00 | 6410000.00 | 6410000.00 | 6410000.00 | 6410000.00 | 6410000.00 | 6410000.00 | 6410000.00 | 6409999.50 | 6409999.50 | 6410000.00 | 6409999.50 | 6409999.00 | n/a | Pa | Inlet s team pres sure |
| TB_OutSteamPress | 5200.00 | 5799.65 | 5952.76 | 6127.13 | 6327.79 | 6561.56 | 6919.54 | 7362.51 | 7907.64 | 8597.21 | 9693.84 | 11301.32 | 14113.57 | 19997.92 | n/a | Pa | Outlet s team pressure |
| TB_IsoValvePos | 1.00 | 1.00 | 1.00 | 1.00 | 1.00 | 1.00 | 1.00 | 1.00 | 1.00 | 1.00 | 1.00 | 1.00 | 1.00 | 1.00 | 0/1 | bool | Isolation Valve Position |
| TB_SpeedCtrlValvePos | 100.00 | 98.35 | 98.47 | 98.60 | 97.75 | 98.93 | 99.14 | 99.40 | 99.71 | 100.12 | 100.66 | 101.41 | 102.52 | 104.42 | 0-100 | % | Speed control valve position (Governor Valve) |
| TB_InSteamFlow | 1490.28 | 1497.05 | 1498.79 | 1500.78 | 1503.07 | 1505.74 | 1508.93 | 1512.78 | 1517.55 | 1523.64 | 1531.72 | 1542.88 | 1559.49 | 1587.61 | n/a | kg/s | Inlet flow |

**Figure 26.** Simulated results (TB–Turbine).

**CC_PumpSpeed Values (75 - 15)**

| TAG | RATED | 75 | 70 | 65 | 60 | 55 | 50 | 45 | 40 | 35 | 30 | 25 | 20 | 15 | RANGE | UNIT | DESCRIPTION |
|---|---|---|---|---|---|---|---|---|---|---|---|---|---|---|---|---|---|
| RX_MeanCoolTemp | 576.75 | 576.75 | 576.75 | 576.75 | 576.75 | 576.75 | 576.75 | 576.75 | 576.75 | 576.75 | 576.75 | 576.75 | 576.75 | 576.75 | n/a | K | Reactor - mean coolant temperature |
| RX_InCoolTemp | 562.94 | 562.94 | 562.94 | 562.94 | 562.95 | 562.94 | 562.94 | 562.94 | 562.94 | 562.94 | 562.94 | 562.94 | 562.94 | 562.94 | n/a | K | Reactor - input coolant temperature |
| RX_OutCoolTemp | 590.62 | 590.61 | 590.62 | 590.61 | 590.60 | 590.62 | 590.61 | 590.62 | 590.62 | 590.62 | 590.61 | 590.61 | 590.62 | 590.61 | n/a | K | Reactor - output coolant temperature |
| RX_CladTemp | 948.28 | 948.28 | 948.28 | 948.28 | 948.28 | 948.28 | 948.28 | 948.28 | 948.28 | 948.28 | 948.28 | 948.28 | 948.28 | 948.28 | n/a | K | Fuel Clad Temperature |
| RX_FuelTemp | 948.28 | 948.28 | 948.28 | 948.28 | 948.28 | 948.28 | 948.28 | 948.28 | 948.28 | 948.28 | 948.28 | 948.28 | 948.28 | 948.28 | n/a | K | Fuel Rod Temperature |
| RX_TotalReac | 0.00 | 0.00 | 0.00 | 0.00 | 0.00 | 0.00 | 0.00 | 0.00 | 0.00 | 0.00 | 0.00 | 0.00 | 0.00 | 0.00 | n/a | $ | Total Reactivity |
| RX_ReactorPower | 100.00 | 100.01 | 100.01 | 100.01 | 100.01 | 100.01 | 100.01 | 100.01 | 100.01 | 100.01 | 100.01 | 100.01 | 100.01 | 100.01 | n/a | % | Reactor Power |
| RX_ReactorPress | 15166000.00 | 15166018.00 | 15166000.00 | 15166015.00 | 15166091.00 | 15166003.00 | 15166000.00 | 15166023.00 | 15166000.00 | 15166000.00 | 15166000.00 | 15166000.00 | 15166031.00 | 15166000.00 | n/a | Pa | Reactor pressure (near the reactor outlet) |
| RX_CL1Press | 15365000.00 | 15365078.00 | 15365000.00 | 15365060.00 | 15365391.00 | 15365011.00 | 15365000.00 | 15365099.00 | 15365000.00 | 15365000.00 | 15365002.00 | 15365002.00 | 15365133.00 | 15365002.00 | n/a | Pa | Reactor cold leg 1 pressure |
| RX_CL2Press | 15365000.00 | 15365078.00 | 15365000.00 | 15365060.00 | 15365391.00 | 15365011.00 | 15365000.00 | 15365098.00 | 15365000.00 | 15365000.00 | 15365002.00 | 15365002.00 | 15365132.00 | 15365002.00 | n/a | Pa | Reactor cold leg 2 pressure |
| RX_CL1Flow | 8801.40 | 8802.70 | 8801.40 | 8802.51 | 8808.03 | 8801.59 | 8801.40 | 8803.04 | 8801.41 | 8801.41 | 8801.43 | 8801.43 | 8803.62 | 8801.43 | n/a | kg/s | Reactor cold leg 1 flow |
| RX_CL2Flow | 8801.40 | 8802.70 | 8801.40 | 8802.51 | 8808.03 | 8801.59 | 8801.40 | 8803.02 | 8801.41 | 8801.41 | 8801.43 | 8801.43 | 8803.60 | 8801.43 | n/a | kg/s | Reactor cold leg 2 flow |
| RX_InOutCoolTemp Avg | 580.00 | 576.80 | 576.80 | 576.80 | 576.80 | 576.80 | 576.80 | 576.80 | 576.80 | 576.80 | 576.80 | 576.80 | 576.80 | 576.80 | n/a | n/a | Reactor Protection System |
| RX_InOutCoolTemp DT | 40.00 | 27.67 | 27.68 | 27.66 | 27.68 | 27.68 | 27.68 | 27.68 | 27.67 | 27.68 | 27.68 | 27.68 | 27.67 | 27.68 | n/a | n/a | Reactor Protection System |

**Figure 27.** Simulated results (RX—Reactor).

From the experiments performed, it was possible to arrive at the following conclusions about the cyber-attack against the condenser cooling pump speed, especially for values below 50:

- The plant's protection system may be triggered, resulting in the interruption of its power generation and consequently in financial losses;
- The significant increase in vapor pressure in the condenser and turbine outlet, to values far above their operating range, could result in physical damage to the equipment, with risks to worker safety. This could also represent a significant financial loss, since in addition to the funds needed to repair or replace the equipment, it would extend the time needed to restore the NPP to its normal activities.

## 5. Intrusion Detection and Defensive Capabilities

Besides demonstrating the ability of the testbed to access realistic cyber-attacks against a nuclear power plant simulator, we would also like to emphasize its potential for conducting studies for the development of intrusion detection techniques and for the possible

automation of this process. Since ICS are deterministic systems, we believe that, for cyber-attacks similar to the one employed, their detection could be done mainly by joint monitoring [42]: network parameters; and operational variables.

To illustrate the network monitoring approach, we have chosen the "Time from request" network parameter in Modbus TCP packets destined for ScadaBR (10.0.0.4). This value measures the response time between the request from the supervisor and the response from the ANS, and can be easily captured by Wireshark, as shown in Figure 28. Next, we extracted about 2000 of these packets, at the point between the supervisor and the Switch (HMI_ScadaBR Ethernet0 to Switch eth3); and used this data for graphical comparison between normal operation and under MITM attack, as depicted in Figures 29 and 30. The approximate average delay (shown in these pictures as colored lines) of about 0.01 seconds is noticeable when we compare the normal response time with the situation under attack.

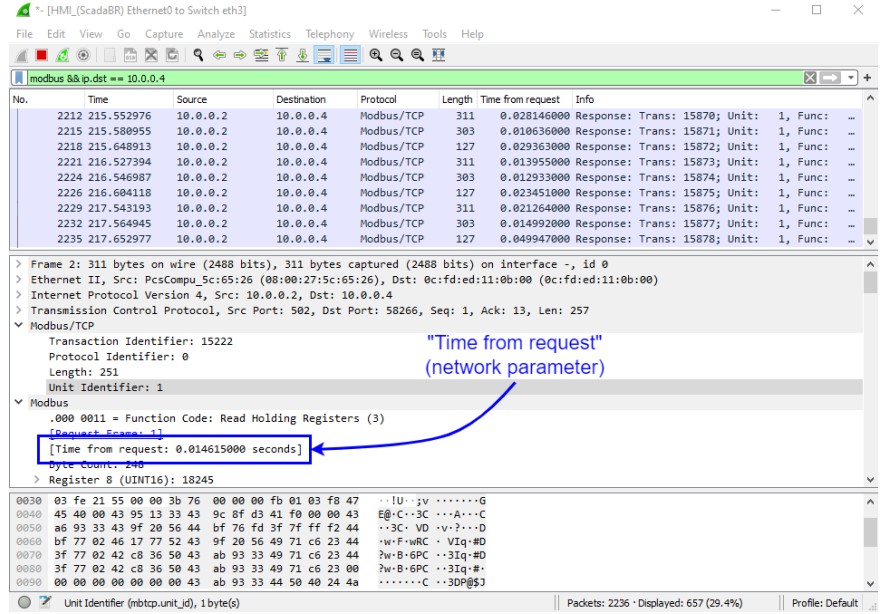

**Figure 28.** Time from request—Wireshark snapshot.

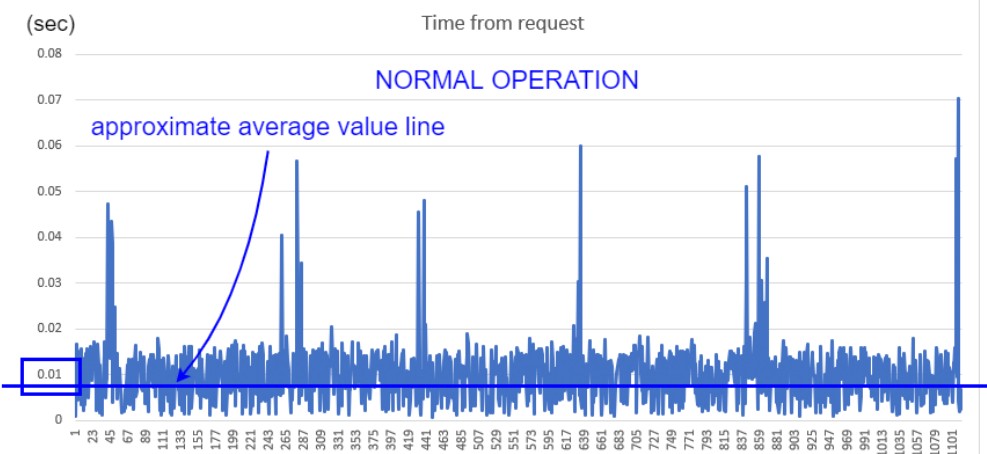

**Figure 29.** TFR comparison—normal.

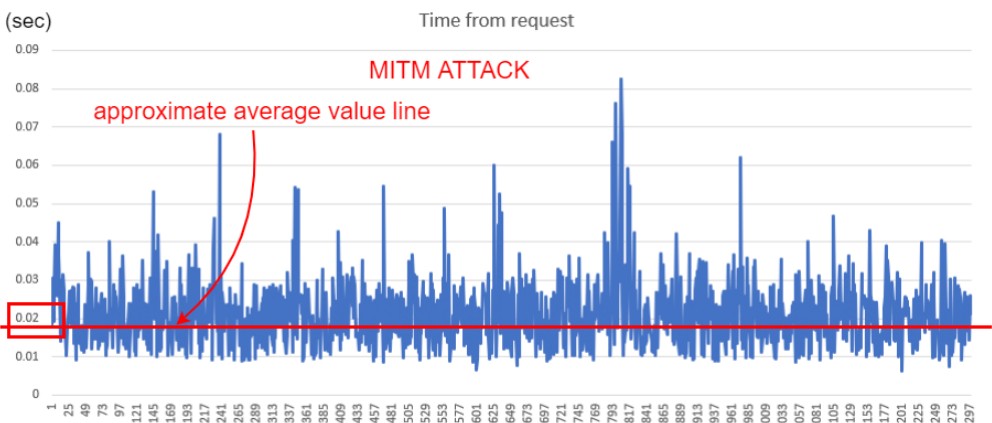

**Figure 30.** TFR comparison—MITM.

To demonstrate the operational variables approach, and guided by the experimental results above described, we chose to relate the variables CC_PumpSpeed and CD_Press. If a pattern for their simultaneous values in a nominal operation situation could be found, this in principle would allow the detection of elaborate cyber-attacks against one of them; such as replay attacks, where the value shown in the HMI corresponds to the oscillations of that variable values in a previous period.

In normal operation with the Main Reactor (RX) power at 100%, the variable CC_PumpSpeed, held at the constant value of 100 by our cyber-attacks, actually oscillates with values close to 102. This can be seen most clearly in the graphs in Figure 31, where the first 1000 or so measurements after the start of normal reactor operation are shown. The graph depicted on the left shows the values for each observation and the graph on the right shows the density or frequency of values associated with the various measurements, for the same data set.

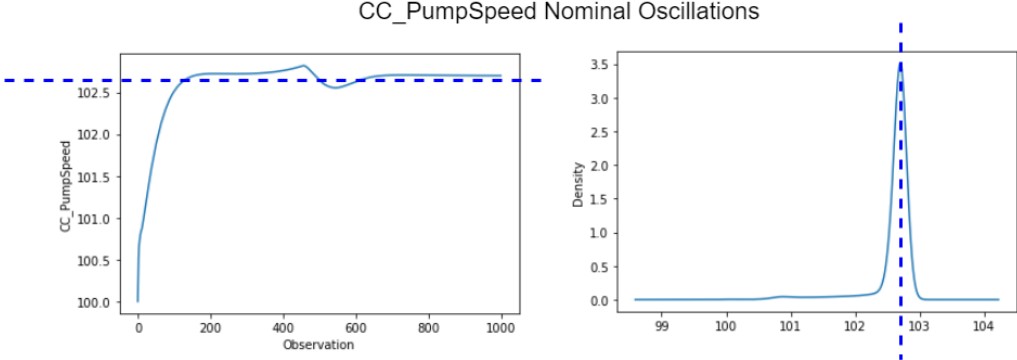

**Figure 31.** CC_PumpSpeed nominal oscillations (around 102).

Furthermore, small operation transients, even in normal operation, are expected and were observed during the simulations. Therefore, we used MySQL Workbench to extract a sample of 600 observations where one of these transients were present, as shown in Figure 32, and thereafter studied the relationship between the values of CC_PumpSpeed and CD_Press in this range.

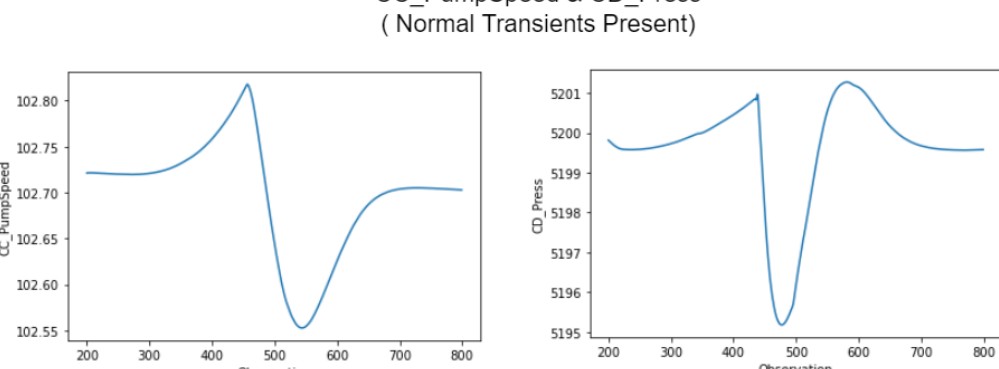

**Figure 32.** CC_PumpSpeed and CD_Press simultaneous transients.

After normalizing these values between 0 and 1, as shown in Figure 33 (a standard preparatory procedure necessary to employ various ML algorithms), we determined the linear correlation between the variables to be very low ($-0.1925$). However, the respective scatter plot clearly displays the formation of an underlying non-linear pattern, as seen in Figure 34. This found locus could presumably be used to detect anomalies. Although mere visual inspection will be insufficient to detect non-linear relationship patterns for more than two chosen variables or features, appropriate mathematical and computational techniques can be employed instead. The same procedures could be expanded in order to train ML algorithms, by including new variables and enlarging the dataset.

Besides the possibility of studying anomaly detection techniques, this testbed also allows the application of a graded approach and defense in depth, by implementing computer security levels and computer security zones, and assessing the technical computer security measures needed to protect facility functions. One example of practical implementation of a defensive computer security architecture (not part of this study), would be the use of a decoupling mechanism between computer security zones containing the PLCs and the supervisory.

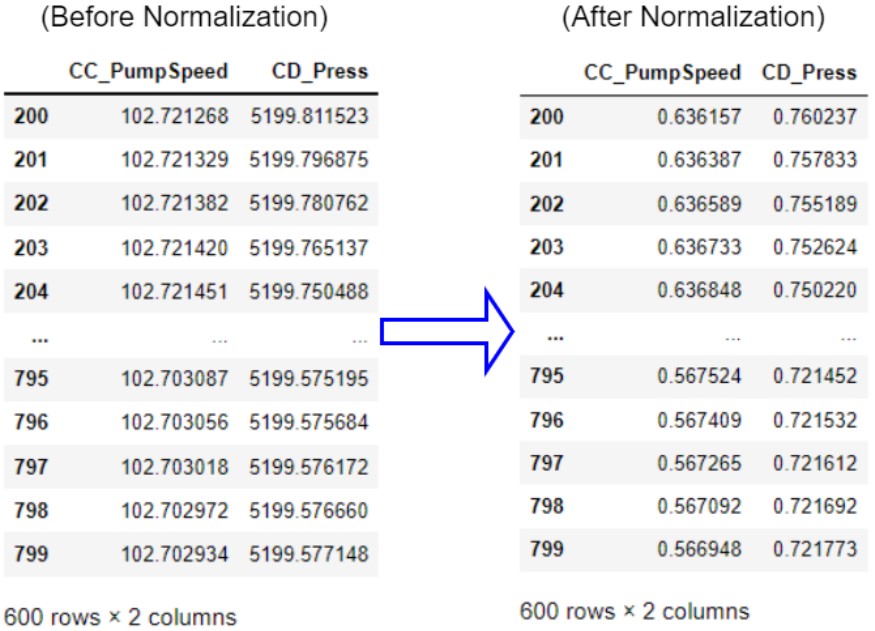

**Figure 33.** CC_PumpSpeed and CD_Press—data normalization.

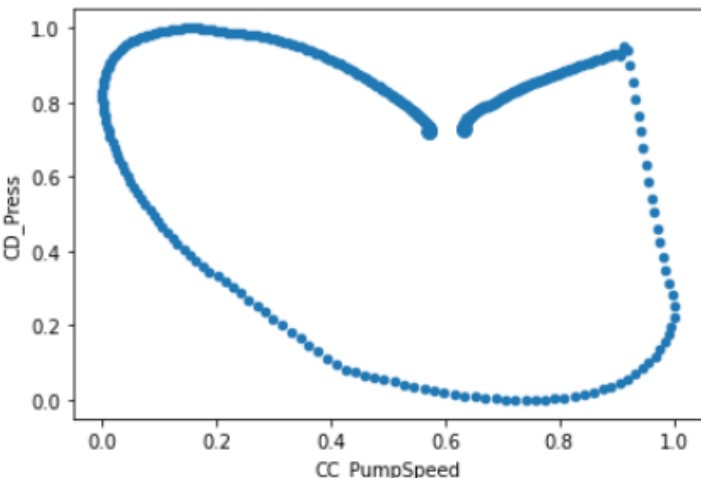

**Figure 34.** CC_PumpSpeed and CD_Press Data—scatter plot.

## 6. Conclusions

In this work, we developed and validated a testbed for conducting cybersecurity assessment in nuclear power plants. The main advantages of this setup are its realism, flexibility, and low cost. It allows the simulation of several cyber-attack scenarios against a simulated NPP communicating with its supervisory system (SCADA/HMI), through the Modbus TCP protocol. It is worth mentioning that this study was carried out based on a simulator of a hypothetical power plant (Asherah NPP) and that vulnerabilities that could lead to a similar attack on existing plants were not exploited. We also showed how it is possible to use this environment to generate the datasets needed for intrusion detection studies, and stated that it allows for implementation of defensive computer security architecture.

For the proposed cyber-attack scenario, the performance of several simulations allowed to demonstrated how to force condenser cooling pump parameters against their nominal operating values is detrimental to the continuous operation of PWR-type NPPs. In particular, the impact assessment points to the risk that cyber-attacks against the condenser cooling pump speed control could result in material and financial damage to the NPP. The situation described also highlights the danger posed by Insiders, endowed with specific knowledge about the inner workings of the NPP ICS and acting within the Air Gap protected area.

We realized that an important limitation of this testbed is the substantial memory load imposed by the employment of several virtual machines in the GNS3 topology, especially in terms of RAM. Thus, more modest computing environments could experience problems when trying to reproduce or expand the conditions described. However, it was possible to perform all the above procedures with a personal computer equipped with an AMD Ryzen 7 3700X 8-Core Processor 3.60 GHz and 32.0 GB of RAM installed. The RAM memories defined in the topology were: 6 GB for the Windows/ANS VM; and 2 GB for each of the Linux VMs.

Several possibilities for future studies are envisioned. Such as:

- Modification of the proposed topology, as by the inclusion of new independent PLCs or network equipment like firewalls, including for the testing of decoupling mechanisms between security zones;
- Choice of different ANS subsystems, for reproducing similar cyber-attack to the one performed in this work;
- Demonstration of different cyber-attacks like DoS and Replay;
- Production of datasets for ML algorithm training, with the goal of developing automated IDS, among others.

**Author Contributions:** Conceptualization, I.B.d.B. and R.T.d.S.J.; Data curation, I.B.d.B.; Formal analysis, I.B.d.B. and R.T.d.S.J.; Funding acquisition, I.B.d.B. and R.T.d.S.J.; Investigation, I.B.d.B.; Methodology, I.B.d.B. and R.T.d.S.J.; Project administration, I.B.d.B. and R.T.d.S.J.; Resources, I.B.d.B. and R.T.d.S.J.; Software, I.B.d.B.; Supervision, R.T.d.S.J.; Validation, I.B.d.B.; Visualization, I.B.d.B.; Writing—original draft, I.B.d.B.; Writing—review and editing, I.B.d.B. and R.T.d.S.J. All authors have read and agreed to the published version of the manuscript.

**Funding:** This research was funded by Agência Brasileira de Inteligência—ABIN—grant number 08/2019.

**Institutional Review Board Statement:** Not applicable.

**Informed Consent Statement:** Not applicable.

**Data Availability Statement:** Not applicable.

**Acknowledgments:** R.T.d.S.J. gratefully acknowledges the support of CNPq grants 465741/2014-2 and 312180/2019-5, the Administrative Council for Economic Defense—CADE grant 08700.000047/2019-14, the National Auditing Department of the Brazilian Health System DENASUS grant 23106.118410/2020-85, the General Attorney of the Union—AGU grant 697.935/2019, the General Attorney's Office for the National Treasure—PGFN grant 23106.148934/2019-67 and the University of Brasilia—UnB COPEI grant 7129. This work was made possible thanks to the support of International Atomic Energy Agency which provided the Asherah NPP Simulator (ANS).

**Conflicts of Interest:** The authors declare no conflict of interest.

## Abbreviations

The following abbreviations are used in this manuscript:

| | |
|---|---|
| ANS | Asherah Nuclear Power Plant Simulator |
| CC | Condenser Cooling (ANS subsystem) |
| CD | Condenser (ANS subsystem) |
| CRP | Coordinated Research Project |
| DCSA | defensive computer security architecture |
| DoS | Denial of Service (type of cyber-attack) |
| FBD | Function Block Diagram (PLC programming language) |
| HIL | hardware-in-the-loop |
| HMI | human-machine interface |
| IAEA | International Atomic Energy Agency |
| IC | instrumentation and control |
| ICS | industrial control systems |
| IDS | intrusion detection systems |
| IF | isolation forest (machine learning algorithm) |
| IL | Instruction List (PLC programming language) |
| IO | input and output |
| IT | information technology |
| LD | Ladder Logic (PLC programming language) |
| MBAP | Modbus Application Protocol |
| MITM | men-in-the-middle (type of cyber-attack) |
| ML | Machine Learning |
| NPP | nuclear power plant |
| OCNN | one-class neural network (machine learning algorithm) |
| OCSVM | one-class support vector machine (machine learning algorithm) |
| OPC UA | Open Platform Communications Unified Architecture |
| OS | operating system |
| OT | operational technology |
| PDU | Protocol Data Unit |
| PLC | programmable logic controllers |
| PWR | Pressurized Water Reactor |

| RDBMS | relational database management system |
| RPS | Reactor Protection System |
| RX | Main Nuclear Reactor (ANS subsystem) |
| SCADA | supervisory control and data acquisition system |
| SCRAM | emergency shutdown |
| SFC | Sequential Function Chart (PLC programming language) |
| ST | Structured Text (PLC programming language) |
| TB | Turbine (ANS subsystem) |
| TLS | Transport Layer Security |
| VM | virtual machines |

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
