# Peer review of "Development of an Open-Source Testbed Based on the Modbus Protocol for Cybersecurity Analysis of Nuclear Power Plants"

_applsci, doi:10.3390/app12157942_

Round 1

Reviewer 1 Report

1. The introduction should indicate the research gaps and research goals. What is the main question addressed by the research?

2. What does it add to the subject area compared with other published material?

3. What specific improvements could the authors consider regarding the methodology?

4. In the Methodology section, the authors should use a flowchart to describe the proceeding flowchart.

5. Please use a standard flowchart to illustrate the process. For example, use an elliptical graph to illustrate the “start” and “end”, use the diamond graph to illustrate the judgment events.

6. The implementation environment is not clear.

Author Response

Dear reviewer,

Thank you very much for reviewing our manuscript and offering constructive suggestions to improve it.

Point 1: The introduction should indicate the research gaps and research goals. What is the main question addressed by the research?

Response 1: As we consider your point indeed important, we modified the introduction to clarify our goals and the research question and why it is relevant to be addressed. Therefore, the following paragraphs are now part of the paper introduction:

“The main question addressed by this research is the design and validation of an easily reproducible and accurate testbed for nuclear power plant (NPP) cybersecurity research. It is important to bring results regarding the protection of such cyber-physical infrastructures because there is crescent concern to attacks against the monitoring and control systems used in real nuclear plants. However, there are inherent risks associated with the safe operation of radioactive materials and high costs involved in suspending a nuclear plant operation for safely testing cyber-attacks and defense measures. This scenario turns the use of nuclear power plant simulations almost unavoidable in these situations. Therefore, presently and in the foreseeable future this question needs to be addressed to comprehend the possible cyber-attacks, their related risks, and to compose adequate protection measures.

As lately increased computing power allows the operation of realistic simulations of nuclear reactors on personal computers, this paper contributes with the design of a robust simulation-based testbed for NPP cybersecurity studies, combining low-cost hardware and software, to enable realistic simulations of the controlled physical processes and the used communications networks. The validation of the proposal rises another paper contribution in the form of a method for simulating cyber-attacks, presenting a case scenario that illustrates how to minimize the cost, difficulty, and complexity NPP cybersecurity analysis, while maximizing the accuracy, reproducibility, and scalability of this type of experimental setup.”

Point 2: What does it add to the subject area compared with other published material?

Response 2: We have been very careful to cite published material related to nuclear power plant testbeds, observing that these works employ either unrealistic/simplistic physical process simulators and/or expensive software and hardware. These characteristics affect the accuracy, reproducibility, and scalability of this type of experimental setup. Compared to the related work, our proposal is intended to better represent realistic physical and logical elements to allow simulation of a nuclear power plant scenario, including a supervisory system and microcontroller PLC actually used for plant operations. Also, differently from related works, we introduce an emulated network environment that allows cyber-attack simulations for the commonly used industrial protocol Modbus, and the analysis of cyber-attacks developed to be used against real ICS. In addition, while some of the related works consider testbed development to be only a preliminary step to achieve diverse specific research goals, we designed our testbed to be of value to the cybersecurity community in general.

Point 3: What specific improvements could the authors consider regarding the methodology?

Response 3: Regarding the methodology, our approach is twofold, since we follow a general design methodology whose results are validated with an embedded cybersecurity analysis method. First, our general approach consists in surveying issues regarding realistic simulations of nuclear power plants and to design and experimentally validate a software testbed for the controlled analysis of cyberattacks against the simulated nuclear plant. The proposal integrates a simulated Modbus/TCP network environment containing basic industrial control elements implemented with open-source software components. Second, we introduce the method and scenario to validate the proposed testbed architecture by performing and analyzing a representative cyberattack in the developed environment. Thus, our methodology is completed by showing the principles for the analysis of other possible cybernetic attacks.

Considering this point and what the reviewer mentioned in point 4 and 5 bellow, a methodological flowchart was inserted in the paper to meet the reviewer's demand.

Point 4:  In the Methodology section, the authors should use a flowchart to describe the proceeding flowchart.

Response 4: Thank you for suggestion. Accordingly, we produced a flowchart to illustrate the process and included it with comments in the section 3 of the paper.

Point 5: Please use a standard flowchart to illustrate the process. For example, use an elliptical graph to illustrate the “start” and “end”, use the diamond graph to illustrate the judgment events.

Response 5: Thank you for suggestion. Accordingly, we produced a flowchart to illustrate the process and included it with comments in the section 3 of the paper.

Point 6: The implementation environment is not clear.

Response 6: we consider it of great importance to provide enough details and descriptions so that other researchers can understand and reproduce the proposed testbed. Therefore, we have verified globally the correction of our several figures, schematics, and tables, as well as code snippets and configurations, and indicated sources for acquiring the software employed.

Please let us know if the above suggested changes are sufficient to make the paper acceptable for publication in the journal Applied Sciences, according to your advice. And do not hesitate to ask us for further corrections and refinements.

Best Regards,

The Authors.

Reviewer 2 Report

The topic is actual and of interest. Title is interesting and clear and appropriate to the paper’s subject matter.

The research area is identified clearly. Keywords are proper.

Research results are presented objectively and logically using 33 figure.

Conclusions are correctly and logically derived from the evidence and/or arguments, presented data.

The authors has chosen the appropriate way to explain themselves using adequate material from accessible and standard sources. The study is based on a source material consisting of 42 sources of a condensed and periodical nature. Source documentation linked to own descriptions and characteristics exhibits competence and takes account of the global achievements dealing with the discussed issues.

I believe that this paper will be of interest to the readership of your journal.

Below are several suggestions that I hope will be helpful in the paper:

Abstract fully and clearly expresses goal, but the main methods and the methodology needs to be explained.

Referencing could be more accurate and unified – sometimes whole name is written, sometimes only the surname.

Author Response

Dear reviewer,

Thank you very much for reviewing our manuscript and offering constructive suggestions to improve it.

Point 1: Abstract fully and clearly expresses goal, but the main methods and the methodology needs to be explained.

Response 1: We agree with you on the need to clarify our methodology in the abstract. Accordingly, we have introduced a sentence announcing our approach, thus resulting in our new version as follows:

Abstract: The possibility of cyber-attacks against critical infrastructure, and in particular nuclear power plants, has prompted several efforts by academia. Many of these works aim to capture the vulnerabilities of the industrial control systems used in these plants through computer simulations and hardware in the loop configurations. However, general results in this area are limited by the cost and diversity of existing commercial equipment and protocols, as well as by the inherent complexity of the nuclear plants. In this context, this work introduces a testbed for the study of cyber-attacks against a realistic simulation of a nuclear power plant. Our approach consists in surveying issues regarding realistic simulations of nuclear power plants and to design and experimentally validate a software testbed for the controlled analysis of cyberattacks against the simulated nuclear plant. The proposal integrates a simulated Modbus/TCP network environment containing basic industrial control elements implemented with open-source software components. We validate the proposed testbed architecture by performing and analyzing a representative cyberattack in the developed environment, thus showing the principles for the analysis of other possible cybernetic attacks.

Additionally, as we consider of great importance to provide enough details and descriptions so that other researchers can understand and reproduce the proposed testbed, we have supplied several figures, schematics, and tables, as well as code snippets and configurations, and indicated sources for acquiring the employed software packages. And, as a result of this review process, we included a methodological flow chart in the beginning of section 3 of our paper.

Point 2: Referencing could be more accurate and unified – sometimes whole name is written, sometimes only the surname.

Response 2: thank you for pointing this out. We revised the bibliography entirely and made the necessary corrections.

Point 3: Does the introduction provide sufficient background and include all relevant references? Can be improved.

Response 3: we noted your concern about the introduction and modified it to extend the background, referring to the “Related Work” section that follows the introduction.

Please let us know if the above suggested changes are sufficient to make the paper acceptable for publication in the journal Applied Sciences, according to your advice. And do not hesitate to ask us for further corrections and refinements.

Best Regards,

The Authors.

Round 2

Reviewer 1 Report

The authors have fixed the previous concerns.